# Regulation by the RNA-binding protein Unkempt at its effector interface

Kriti Shah [1,2,9], Shiyang He [1,2,9], David J. Turner[3,9], Joshua Corbo [3,8], Khadija Rebbani [3], Daniel Dominguez[4], Joseph M. Bateman [5], Sihem Cheloufi [1,2,6], Cátia Igreja [7], Eugene Valkov [3] ✉ & Jernej Murn [1,2] ✉

How RNA-binding proteins (RBPs) convey regulatory instructions to the core effectors of RNA processing is unclear. Here, we document the existence and functions of a multivalent RBP–effector interface. We show that the effector interface of a conserved RBP with an essential role in metazoan development, Unkempt, is mediated by a novel type of 'dual-purpose' peptide motifs that can contact two different surfaces of interacting proteins. Unexpectedly, we find that the multivalent contacts do not merely serve effector recruitment but are required for the accuracy of RNA recognition by Unkempt. Systems analyses reveal that multivalent RBP–effector contacts can repurpose the principal activity of an effector for a different function, as we demonstrate for the reuse of the central eukaryotic mRNA decay factor CCR4-NOT in translational control. Our study establishes the molecular assembly and functional principles of an RBP–effector interface.

RNA processing is executed by a diverse set of effector proteins and protein complexes that specialize in facilitating a particular molecular event during the lifetime of an RNA. Although several effectors may act on RNA with little specificity, effector activities are often regulated to affect certain transcripts or their parts more than others[1,2]. Specificity in RNA processing is essential for a range of key cellular functions, including cellular differentiation, timely responses to immune signaling, or synaptic plasticity[3-5]. Critical to regulated RNA processing are RNA-binding proteins (RBPs) that can, by interacting with both the RNA and the effector, divert effector activity toward specific, RBP-targeted transcripts[1,2,6].

One major unsolved problem associated with regulated RNA processing is a fragmented understanding of RBP–effector interactions. In particular, it is unclear how interactions of different RBPs with a given effector can direct different modes of RNA processing. For instance, regulatory RBPs that bind to the same relative positions on RNA can, through recruitment of spliceosomal components, either promote or antagonize the maturation of a functional spliceosome, resulting in opposing effects on RNA splicing[7]. Likewise, RBP-dependent recruitment of the CCR4-NOT effector complex can favor either decay or translational repression of targeted transcripts to serve different cellular functions[8-12]. It is unknown whether RBP–effector interactions also contribute to the RNA recognition by the RBP, which could, on its own, influence both the specificity and the mode of RNA processing[13,14].

An emerging theme is that RBP–effector interactions are often mediated by a short linear motif (SLiM) embedded within RBP's intrinsically disordered region (IDR), which supports direct contact with a structured domain of an effector protein[2]. A few essential residues in these SLiMs provide specificity in transient interactions[15]. However, functional studies of SLiMs and the derivation of generalizable principles are complicated by the evolutionary plasticity of SLiMs at the sequence level.

[1]Department of Biochemistry, University of California, Riverside, 3401 Watkins Drive, Boyce Hall, Riverside, CA 92521, USA. [2]Center for RNA Biology and Medicine, 900 University Ave, Riverside, CA 92521, USA. [3]RNA Biology Laboratory, Center for Cancer Research, National Cancer Institute, Frederick, MD 21702, USA. [4]Department of Pharmacology, University of North Carolina at Chapel Hill, Chapel Hill, NC 27599, USA. [5]Maurice Wohl Clinical Neuroscience Institute, King's College London, 5 Cutcombe Road, SE5 9RX London, UK. [6]Stem Cell Center, University of California, Riverside, 900 University Ave, Riverside, CA 92521, USA. [7]Department for Integrative Evolutionary Biology, Max-Planck-Ring 9, D-72076 Tübingen, Germany. [8]Present address: Department of Molecular and Cellular Biology, Harvard University, 52 Oxford Street, Cambridge, MA 02138, USA. [9]These authors contributed equally: Kriti Shah, Shiyang He, David J. Turner. ✉e-mail: eugene.valkov@nih.gov; jernej.murn@ucr.edu

Here, we study a developmentally essential RBP–effector interface via its control of RNA processing. As a model system, we use the sequence-specific RBP Unkempt (UNK), taking advantage of its distinct molecular features and a clear cellular phenotype that we utilize as sensitive functional readouts[16,17]. This includes UNK's strict requirement for a specific RNA-binding motif, its potent transcriptional and translational activities, as well as its unique capacity to induce a bipolar cellular morphology, an activity that is required during early neurogenesis and that can be recapitulated in non-neuronal cells (Fig. 1A, B)[16–18]. We identified numerous RBP–effector contacts maintained via IDR-embedded SLiMs and arranged via RBP dimerization. Interactions of UNK with each of its key effectors, CCR4-NOT and poly(A)-binding protein (PABPC), substantially contribute to the recognition of UNK's RNA-binding motif, with PABPC additionally playing a dominant role in positioning UNK on mRNA and with CCR4-NOT mediating target-specific translational repression. Our findings define an RBP–effector interface and elucidate its central role in specifying the regulatory function of an RBP.

## Results

### Unkempt's intrinsically disordered region is a hub of regulatory activity

UNK is a cytoplasmic, translationally active RBP with a critical role in the development of the nervous system[16,19–22]. Prior to its identification as an RBP, however, we discovered that UNK was transcriptionally active in a dual-luciferase reporter assay (Supplementary Figs. 1, 2A, B). We mapped UNK's transcriptional activity to an extended and conserved IDR (Fig. 1C, Supplementary Fig. 2C, D, Supplementary Data 1). Deleting the entire IDR or its portions silenced transcriptional activation in this assay (Supplementary Fig. 2D). Moreover, we found that transcriptionally more potent UNK mutants induced stronger morphological transformation of cells, suggesting that IDR is required for UNK function (Fig. 1A, B, Supplementary Fig. 2E, F).

In a further screen of IDR mutants, we defined two shortest active regions, minN, and minC (Supplementary Fig. 3A, B, Supplementary Data 1). Strikingly, substituting L522 in minN or two residues in minC (W622 and F625) to alanines completely silenced the activities of either region (Fig. 1C, Supplementary Fig. 3A). A combined mutant, UNK$_{3M}$, encompassing L522A/W622A/F625A, silenced not only IDR but also the full-length UNK protein (Supplementary Fig. 3C, D). Notably, as seen with the deletion mutants (Supplementary Fig. 2F), we observed positive correlation between the cell-polarizing activities of UNK

residue mutants and their transcriptional activities, with UNK$_{3M}$ as the minimal mutant that failed to elicit cellular polarization (Supplementary Fig. 3E, F).

Because UNK is nearly exclusively cytoplasmic, its transcriptional activity was unexpected (Supplementary Fig. 1B, C)[16,19,20]. To dissect a potential role of UNK in transcription, we performed ChIP-seq analyses of endogenous or ectopic UNK, RNA-guided recruitment of dCas9 fusions with UNK or its IDR to loci of endogenous genes to induce their transcription, and mass spectrometry analyses of affinity-purified nuclear protein complexes of UNK to identify any chromatin-associated interactors. None of these analyses suggested a transcriptional activity for UNK, although we cannot rule out its biological relevance. We further pursued the function of IDR due to its strict requirement for the morphogenetic activity of UNK (Supplementary Figs. 2E, F, 3E, F).

### CCR4-NOT and PABPC are critical effectors of Unkempt

Proximity-dependent biotinylation (BioID) analysis identified several hundred UNK interactors in cells (Fig. 2A, Supplementary Data 2)[23]. To assess their impact on UNK function, we compared compositions of complexes formed by the wild-type (WT) UNK (UNK$_{WT}$) or the inactive UNK$_{3M}$ by mass spectrometry (Fig. 2B, Supplementary Data 2). We observed a major difference in the association with the CCR4-NOT complex subunits; whereas all CCR4-NOT subunits were readily detected in the UNK$_{WT}$ complex, they were absent in the UNK$_{3M}$ complex (Fig. 2B, Supplementary Fig. 4A, Supplementary Data 2). This was confirmed by a co-IP/western analysis that further pointed to the contribution of each of the three mutated residues to the interaction between UNK and CCR4-NOT (Fig. 2C). Thus, the 3M mutation that renders UNK morphogenetically inactive also specifically disrupts its association with the CCR4-NOT complex.

To validate these results genetically, we tested the morphogenetic capacity of UNK$_{WT}$ in HeLa cells following siRNA-mediated knockdown (KD) of individual CCR4-NOT subunits. Interestingly, suppressing the deadenylase activity of CCR4-NOT via a simultaneous KD of CNOT7 and CNOT8 only minimally impacted cell polarization, whereas KD of CNOT9 substantially impaired the capacity of UNK$_{WT}$ to transform cellular morphology (Fig. 2D, E)[24]. We confirmed the requirement for CNOT9 in *CNOT9*-null cells that were nearly fully resistant to the morphogenetic activity of UNK$_{WT}$ (Fig. 2F, G). Furthermore, the absence of CNOT9 substantially reduced the interaction of UNK with

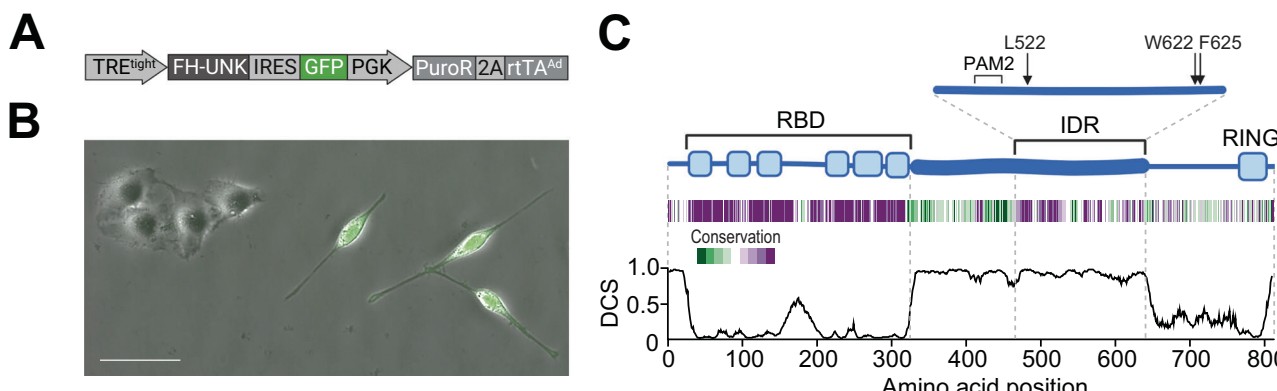

**Fig. 1 | Morphogenetic activity of Unkempt and the identified activity-linked features in its IDR. A** Inducible system for UNK-driven cell morphogenesis. Shown is an all-in-one variant of the previously reported system used in this study (see "Methods" section)[16,17]. **B** HeLa cells are incubated with doxycycline (Dox) for 48 h after which the morphology of GFP-expressing cells is evaluated (see "Methods" section). A representative view from *n* = 6 independent experiments is shown. Scale bar, 50 μm. **C** Domain map of UNK (blue) indicating its RNA-binding domain (RBD),

intrinsically disordered region (IDR), and a RING finger domain. Amino acid conservation (green, least conserved; purple, most conserved position) and disorder confidence profile of UNK are shown below the map (see "Methods" section). IDR is shown enlarged above the domain map with the locations of identified activity-linked residues (L522, W622, F625) and the predicted PABPC-binding motif (PAM2) indicated. DCS, disorder confidence score.

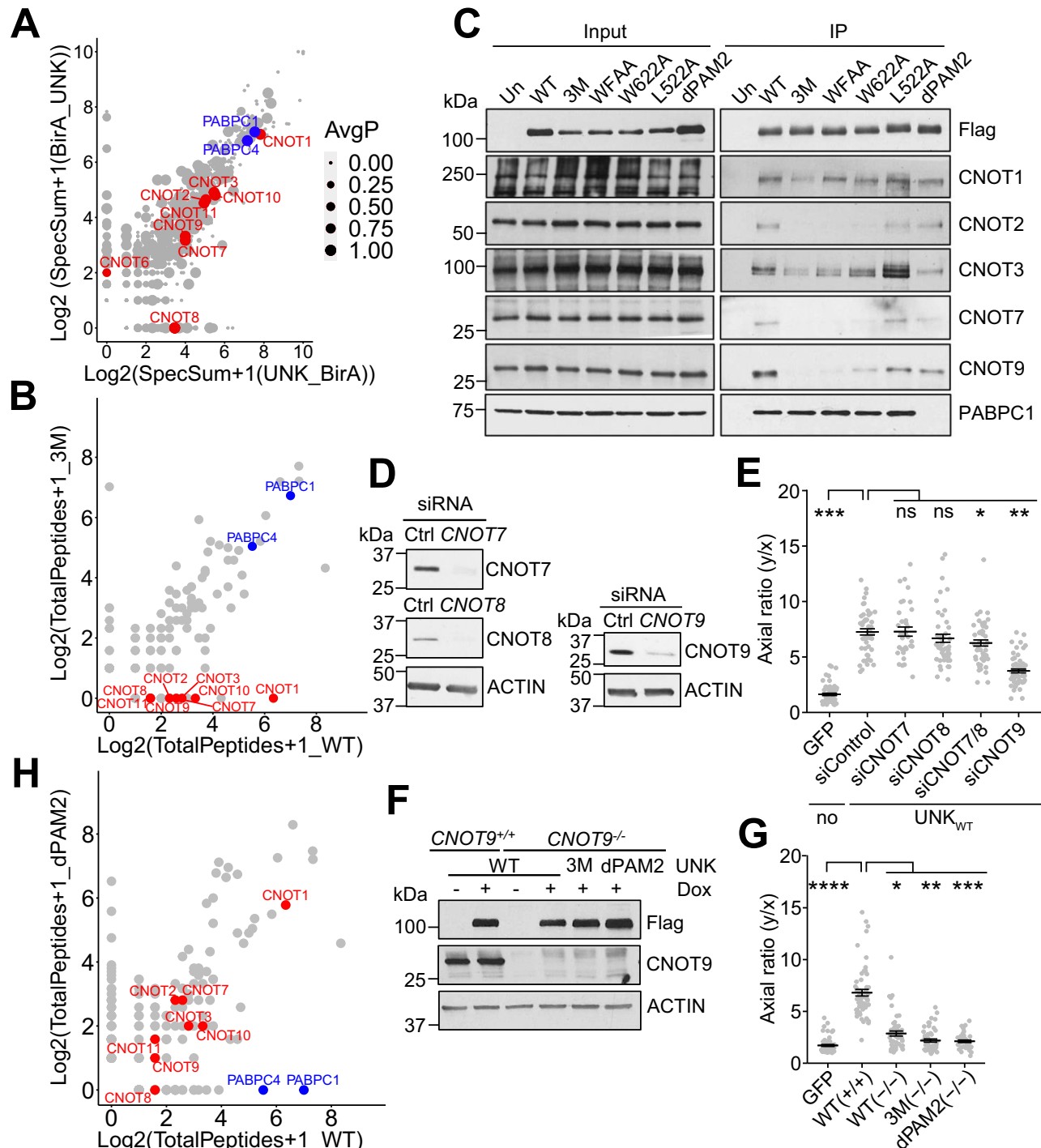

**Fig. 2 | The CCR4-NOT complex is a critical effector of Unkempt. A** Result of a BioID analysis showing the comparison of average spectral counts of peptides derived from each protein identified as an interactor of UNK with an abortive biotin ligase (BirA) fused to either N-terminus (BirA_UNK) or C-terminus (UNK_BirA) of UNK. The diameter of each interactor is proportional to the average probability of interaction (AvgP). Reanalysis of data from Youn et al. [23]. **B** Results of mass spectrometry analyses comparing total peptide counts detected in protein complexes of UNK$_{WT}$ and UNK$_{3M}$. **C** Co-IP of endogenous CCR4-NOT subunits and PABPC1 with UNK from lysates of HeLa cells inducibly expressing the indicated Flag-HA-tagged UNK mutants. Precipitated proteins were detected by western blot analysis. WFAA, UNK with W622A and F625A mutations. **D** Knockdown of the indicated CCR4-NOT subunits at 48 h after transfection of their targeting siRNAs in cells inducibly co-expressing UNK$_{WT}$ and GFP. **E** Morphologies of siRNA-transfected cells were

quantified at 48 h of incubation with Dox ($n$ = between 35 and 59 GFP-expressing cells per cell line). **F** Western analysis of WT (CNOT9$^{+/+}$) or CNOT9 knockout cells (CNOT9$^{-/-}$) inducibly expressing Flag-HA-tagged UNK$_{WT}$, UNK$_{3M}$, or UNK$_{dPAM2}$ at 24 h of induction with Dox. **G** As in (**F**), morphologies of cells expressing GFP alone or co-expressing Flag-HA-tagged UNK$_{WT}$, UNK$_{3M}$, or UNK$_{dPAM2}$ and GFP in CNOT9$^{+/+}$ or CNOT9$^{-/-}$ cells were quantified at 48 h of incubation with Dox ($n$ = between 39 and 59 GFP-expressing cells per cell line). **H** Results of mass spectrometry analyses comparing total peptide counts detected in protein complexes of UNK$_{WT}$ and UNK$_{dPAM2}$. Blots in **C** ($n$ = 3), **D** ($n$ = 3), and **F** ($n$ = 2) are representative of biologically independent repeats. Data in (**E**) and (**G**) are presented as mean ± SD. Statistical significance was determined using Student's two-tailed $t$-test with *$p$ = 0.015, **$p$ = 8.1 × 10$^{-9}$, ***$p$ = 1.6 × 10$^{-33}$ in (**E**) and *$p$ = 1.5 × 10$^{-16}$, **$p$ = 9.4 × 10$^{-22}$, ***$p$ = 5.7 × 10$^{-22}$, ****$p$ = 1.0 × 10$^{-23}$ in (**G**). ns not significant.

other CCR4-NOT subunits (Supplementary Fig. 4B), implicating CNOT9 as the principal binding site for UNK on CCR4-NOT.

The 3M mutation did not affect interactions with two of the strongest binding partners of UNK, the cytoplasmic poly(A)-binding proteins PABPC1 and PABPC4 (collectively termed PABPC; Fig. 2A–C, Supplementary Fig. 4A)[2,25]. In the UNK protein sequence, we identified a putative SLiM, known as a PAM2 motif, found in diverse proteins that bind to the MLLE domain of PABPC (Fig. 1C, Supplementary Fig. 3A, 4C)[26,27]. We deleted the identified PAM2-like motif from the full-length UNK and named the resulting mutant protein UNK$_{dPAM2}$. As assessed by mass spectrometry and confirmed by co-IP/western analysis, UNK$_{dPAM2}$ bound the CCR4-NOT complex, akin to UNK$_{WT}$, but did not interact with PABPC (Fig. 2C, H, Supplementary Fig. 4A, Supplementary Data 2). Notably, UNK$_{dPAM2}$ exerted weaker transcriptional and morphogenetic activities than UNK$_{WT}$ (Supplementary Fig. 3D–F). Together, these findings identify CCR4-NOT and PABPC as key effectors of UNK and further suggest that the role of CCR4-NOT is independent of its deadenylase activity.

## Unkempt interacts directly with the NOT and NOT9 modules of the CCR4-NOT complex

To investigate the directness of interactions between UNK and CCR4-NOT, we carried out in vitro pull-down assays with recombinant full-length UNK (UNK$_{FULL}$) that was immobilized on beads and incubated with CCR4-NOT subcomplexes (modules) reconstituted from purified recombinant proteins (Fig. 3A)[28]. UNK$_{FULL}$ bound specifically to the NOT9 and NOT modules

but not the catalytic or NOT10/11 modules (Fig. 3B). Both the NOT9 and NOT modules were also directly bound by the functionally essential segment, UNK$_{IDR}$ (Figs. 1C, 3C). However, UNK$_{IDR}$ with the 3M mutation only inefficiently pulled down the NOT module and did not bind the NOT9 module (Fig. 3C). A mutational analysis of the individual 3M residues further pointed to a major role for W622 and minor contributions of L522 and F625 in supporting direct contact with the CCR4-NOT modules (Fig. 3C).

We then used AlphaFold employing the rigorous approach proposed by Conti and co-workers to generate structure predictions of UNK$_{IDR}$ in complex with the NOT9 module (Fig. 3D–G)[29–31]. Interestingly, these predictions suggested that a segment of UNK$_{IDR}$ folds into a helix (residues 507–537; henceforth SLiM 1) that binds across the CNOT9 concave surface (Fig. 3D, E), which serves as a protein−protein interaction site for several other factors[32], whereas the sole tryptophan residue within UNK$_{IDR}$, W622, inserts into either of the defined tryptophan (W)-binding pockets on the convex surface of CNOT9 (Fig. 3F, G). We denote W622 and its neighboring residues in contact with CCR4-NOT (residues 617–625) as SLiM 2.

Intriguingly, AlphaFold predicted that SLiM 1 may also mediate an interaction between UNK$_{IDR}$ and the NOT module (Fig. 3H, I). Specifically, SLiM 1 was predicted again to fold into a helix and bind to a conserved hydrophobic pocket on the surface of the CNOT1 C-terminal domain (Fig. 3H, I). Although these predictions did not point to a clear SLiM 2 binding site on the NOT module, the pull-down assays showing that mutating either W622 or F625 reduces the binding of UNK$_{IDR}$ with the NOT module suggested such interaction (Fig. 3C).

We first tested whether UNK$_{IDR}$ interacts with NOT9 W-pocket mutants to validate the predicted interfaces. The double W-pocket mutant (NOT9 M3) showed the most impaired interaction with UNK$_{IDR}$ (Fig. 3J). Next, we substituted three hydrophobic residues (V511, I515, and L522) within SLiM 1 to glutamates; this mutant UNK$_{IDR}$ less efficiently recruited either the NOT9 or NOT module and was completely unable to interact with the NOT9 double W-pocket mutant (Fig. 3K). These results support a multivalent mode of UNK$_{IDR}$ interaction with CNOT9 and support the observation that UNK uses the same motifs to bind NOT9 and NOT modules of the CCR4-NOT complex.

## Unkempt binds its effectors as a dimer stabilized by a conserved coiled coil

UNK contains a region with a distinct heptad repeat pattern of a coiled-coil motif (residues 643–767; Supplementary Fig. 5A). To investigate the possible structural role of this motif, we used Alpha-Fold to generate structure predictions for UNK$_{FULL}$ and a C-terminal fragment, residues 637–810, termed UNK$_{C}$. Both predictions revealed two parallel coiled coils stabilizing a putative dimer (Fig. 4A, Supplementary Fig. 5B). To see whether UNK indeed dimerized in solution, we measured the molecular weight of purified UNK$_{FULL}$ and UNK$_{C}$ by mass photometry and confirmed both as exclusive dimers (Fig. 4B).

To validate the dimerization interface, we substituted hydrophobic residues in $d$ positions of the heptad repeats that form coiled-coil motifs for glutamates, generating UNK$_{E8}$ and UNK$_{E6}$, with the latter having substitutions only in the more extended coiled-coil motif (Fig. 4A). These substitutions placed negatively charged residues opposite each other in the coiled coil, leading to electrostatic repulsion and destabilization of the interface. UNK$_{E6}$ was a mixed species of monomers and dimers, while UNK$_{E8}$ was an exclusive monomer, suggesting that both coiled-coil motifs are essential for dimer stability (Fig. 4C).

UNK$_{FULL}$ efficiently pulled down abridged recombinant CCR4-NOT subcomplexes containing both the NOT9 and NOT modules, either the four-subunit CNOT1/2/3/9 or six-subunit CNOT$_{MINI}$ complex, consistent with direct, stable binding (Supplementary Fig. 5C, D)[28]. To determine the stoichiometry of binding, we measured the mass of a reconstituted complex of UNK$_{FULL}$ with CNOT$_{MINI}$, revealing that two copies bind one CNOT$_{MINI}$ (Fig. 4D). Although UNK$_{FULL}$ also pulled down recombinant PABPC1, we could not determine the stoichiometry, suggesting that UNK does not bind PABPC1 as stably as CCR4-NOT (Supplementary Fig. 5D). However, including CNOT1/2/3/9 in the binding reaction had no apparent effect on the pull-down of PABPC1 (Supplementary Fig. 5D), suggesting that PABPC and CCR4-NOT may interact with UNK independently.

Given the capacity of both SLiM 1 and SLiM 2 to bind different CCR4-NOT subunits, dimerization may enhance the stability of the UNK−CCR4-NOT interface through avidity effects. We asked whether dimerization might be important for UNK's cellular function. Strikingly, the monomeric UNK$_{E8}$ failed to alter cell morphology analogous to the UNK$_{3M}$ phenotype (Fig. 4E, Supplementary Fig. 3E). Thus, the IDR-embedded SLiMs and homodimerization are essential for UNK's morphogenetic activity. Notably, the predicted conservation of UNK's propensity to dimerize (Supplementary Fig. 5E) and form SLiM-mediated contacts with CCR4-NOT (Supplementary Fig. 5F), suggest evolutionary constraints that may support the observed conservation of the morphogenetic activity of UNK[17].

## Effector interactions regulate RNA sequence recognition by Unkempt

UNK's consensus RNA recognition sequence is specified by its two CCCH-type zinc finger clusters and consists of a UAG motif upstream of a U/A-rich trimer[16,17]. However, as often observed for sequence-specific RBPs, less than a quarter of the predicted mRNA-binding sites are occupied by UNK in cells and the majority of the observed binding sites do not contain the consensus recognition sequence[16,33]. This led us to ask whether UNK−effector interface may function as an auxiliary determinant of RNA binding by UNK.

To test this, we performed crosslinking and immunoprecipitation using an improved protocol (iCLIP2) for UNK$_{WT}$, UNK$_{dPAM2}$, and UNK$_{3M}$[34]. Analysis of the UNK$_{WT}$ dataset revealed several thousand mRNA targets, a substantial increase over the initially annotated pool of UNK-bound messages (Supplementary Fig. 6A–D, Supplementary Data 3)[16]. In line with the earlier study, we found that UNK-binding sites distributed broadly over the coding regions and 3′UTRs of mRNAs and

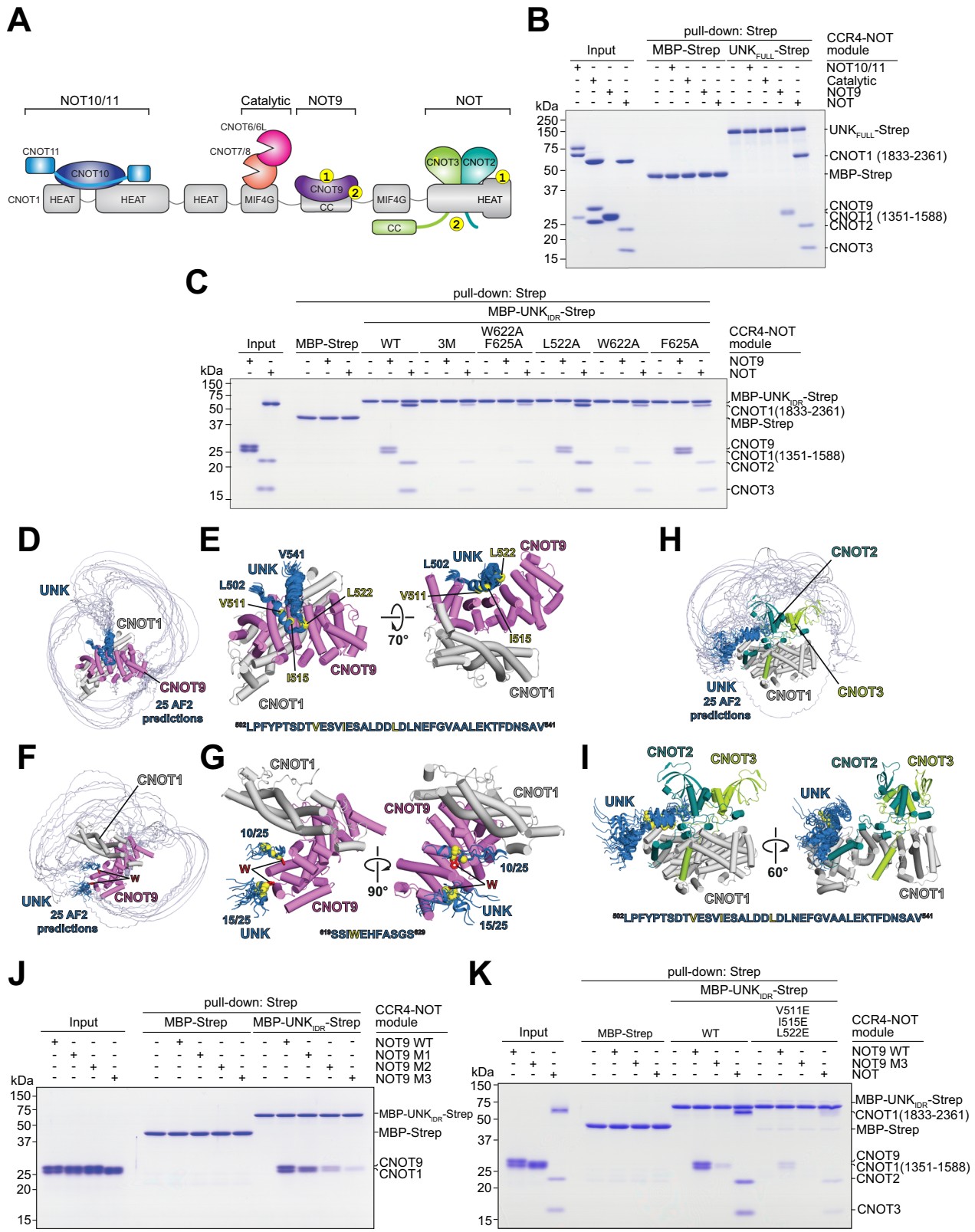

were enriched in UAG and U/A-rich motifs just up- and downstream of the binding peak, respectively (Fig. 5A, Supplementary Fig. 6E, F). Curiously, a similar analysis of UNK$_{dPAM2}$ and UNK$_{3M}$ revealed an altered RNA-binding pattern with a significantly weaker enrichment of the critical UAG motif and with changes in the position-specific representation of several U/A-rich motifs (Fig. 5A, B, Supplementary Data 3).

Although both mutants retained the broad mRNA-targeting potential of UNK$_{WT}$ (Supplementary Fig. 6G, Supplementary Data 3), a consideration of individual RNA-binding events pointed to clear differences between either mutant and UNK$_{WT}$ (Fig. 5C). Interestingly, these differences were less apparent in a mutant-to-mutant comparison, much like the relative similarity in the consensus sequence recognition by UNK$_{dPAM2}$ and UNK$_{3M}$ (Fig. 5A–C). Inspection of

**Fig. 3 | Definition of interactions between Unkempt and CCR4-NOT.**
**A** Schematic representation of CCR4-NOT. Yellow circles indicate points of contact with UNK_IDR identified in this study. The numbering of contacts indicates SLiM 1 or SLiM 2 binding sites. Adapted from Raisch et al. [28] under permission provided by a Creative Commons Attribution 4.0 International License. **B** Pull-down assays with recombinant UNK_FULL tagged with the StrepII (Strep) affinity tag upon incubation with different CCR4-NOT modules ($n = 3$). **C** As in (**B**) but with mutants of UNK_IDR constructs fused to MBP and Strep after incubation with the NOT9 or the NOT module ($n = 3$). **D** Twenty-five AlphaFold predictions of interfaces of UNK_IDR interacting with the NOT9 module. The predictions are aligned on the CNOT9/CNOT1 heterodimer[51]. The region of UNK_IDR where the predictions converged is in dark blue. **E** The converged region of UNK_IDR bound on the concave surface of CNOT9/CNOT1. **F** The same 25 predictions as in (**D**) but oriented to show the tryptophan (W)-binding pockets of CNOT9. **G** All 25

predictions of the converged region of UNK_IDR close to the W-binding pockets of CNOT9. **H** Twenty-five AlphaFold predictions of UNK_IDR interacting with the NOT module. The predictions are aligned on the CNOT1/CNOT2/CNOT3 heterotrimer[58]. The region of UNK_IDR where predictions converged is in dark blue. **I** The converged region of UNK_IDR bound on the surface of CNOT1. **J** Pull-down of WT or M1-M3 mutants of the NOT9 module by MBP-UNK_IDR-Strep. Residues in CNOT9 mutated to alanines in M1 (Y203 and R244) line the W-pocket 1, and those mutated in M2 (R205 and H208) line the W-pocket 2[51]. All residues (Y203, R205, H208, and R244) were mutated in M3 ($n = 2$). **K** Pull-down of the WT or M3 NOT9 module or the NOT module by WT MBP-UNK_IDR-Strep or its mutant with key residues in SLiM 1 (V511, I515, L522) substituted with glutamic acid. In (**E**), (**G**), and (**I**), the C-alpha atoms of the key interacting residues are shown as yellow spheres. The sequences of the converged region are shown with the key interacting residues in yellow.

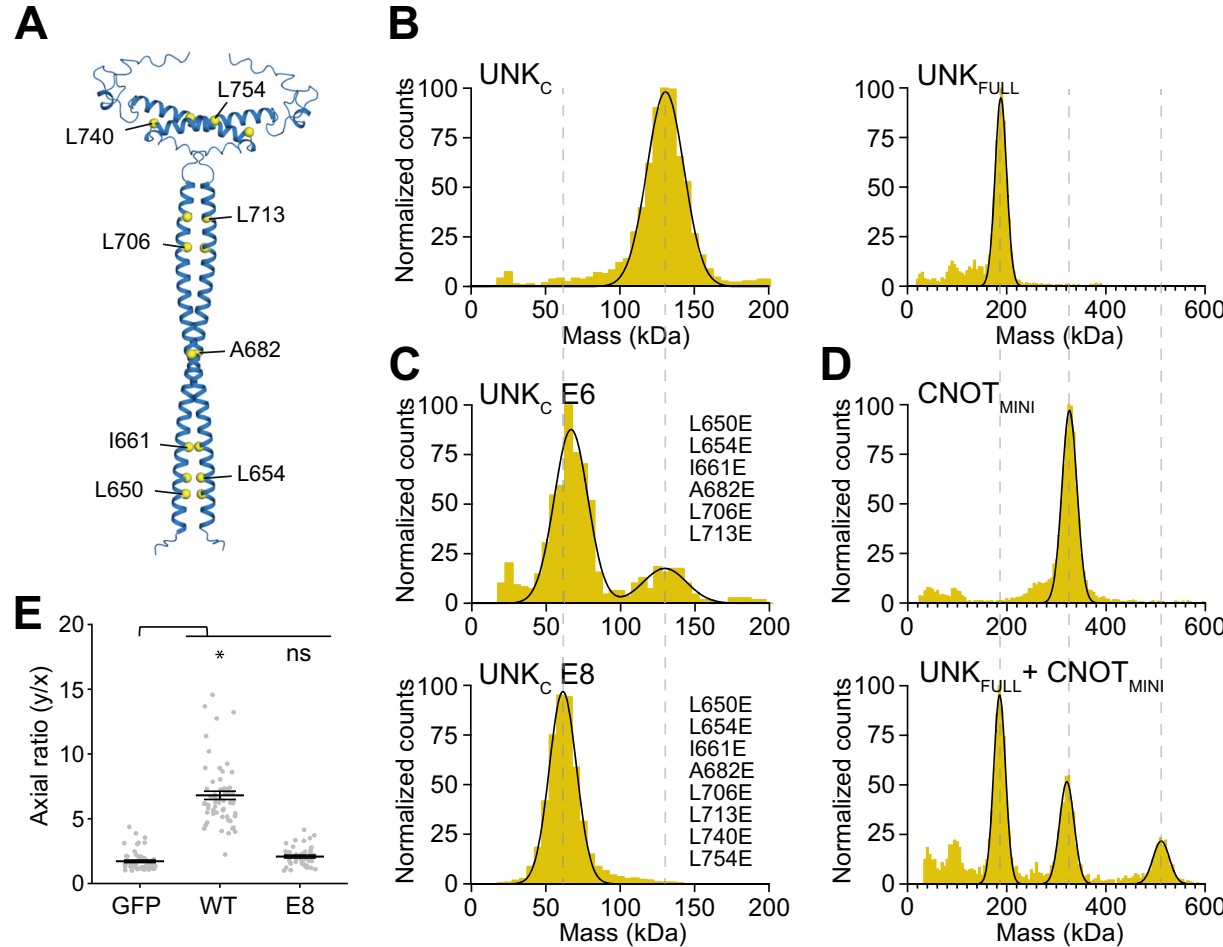

**Fig. 4 | A critical role for dimerization of Unkempt. A** A representative AlphaFold prediction of the coiled-coil homodimer formed by two copies of the C-terminal segment of UNK (UNK_C; residues 637–810). All 25 predictions were identical, bar some minor variations in the C-terminal loop regions. The C-alpha atoms of the key hydrophobic residues stabilizing the coiled coil are shown as yellow spheres. Predictions of the homodimer full-length UNK protein are shown in Supplementary Fig. 5B. **B–D** Mass photometry analyses of Strep-tagged UNK_C or UNK_FULL (**B**), UNK_C E6 or UNK_C E8 (**C**), CNOT_MINI or in complex with UNK_FULL (**D**) ($n = 2$). Mutations in UNK_C E6 and UNK_C E8 are listed in (**C**). The calculated molecular weights are

62.5 kDa for UNK_C monomer, 125 kDa for UNK_C dimer, 90.5 kDa for UNK_FULL monomer, and 181 kDa for UNK_FULL dimer. The observed mean molecular weights are 126 kDa for UNK_C, 62 kDa and 125 kDa for UNK_C E6, 58 kDa for UNK_C E8, 189 kDa for UNK_FULL, 326 kDa for CNOT_MINI, and 186 kDa, 321 kDa, and 512 kDa for the mixture of UNK_FULL and CNOT_MINI. **E** Morphologies of cells inducibly co-expressing UNK_WT or UNK_E8 and GFP compared to GFP-only expressing cells at 48 h of incubation with Dox ($n$ = between 44 and 59 GFP-expressing cells per cell line). Data are presented as mean ± SD. Statistical significance was determined using Student's two-tailed t-test with *$p = 1.0 \times 10^{-23}$; ns not significant.

individual target transcripts revealed weakened targeting of UAG-containing sites by the mutants compared to UNK_WT with concomitant emergence of UAG-less 'satellite' peaks (Fig. 5D, E). Taken together, effector interactions distinctly contribute to the accuracy of RNA sequence recognition by UNK in cells.

## PABPC controls the distribution of Unkempt on mRNA
Both CCR4-NOT and PABPC are thought to locate largely at the 3' ends of mRNAs, although the precise positions of the mammalian CCR4-NOT have not been determined[25,35–37]. We asked how effector localization may affect the distribution of UNK on mRNAs. Strikingly,

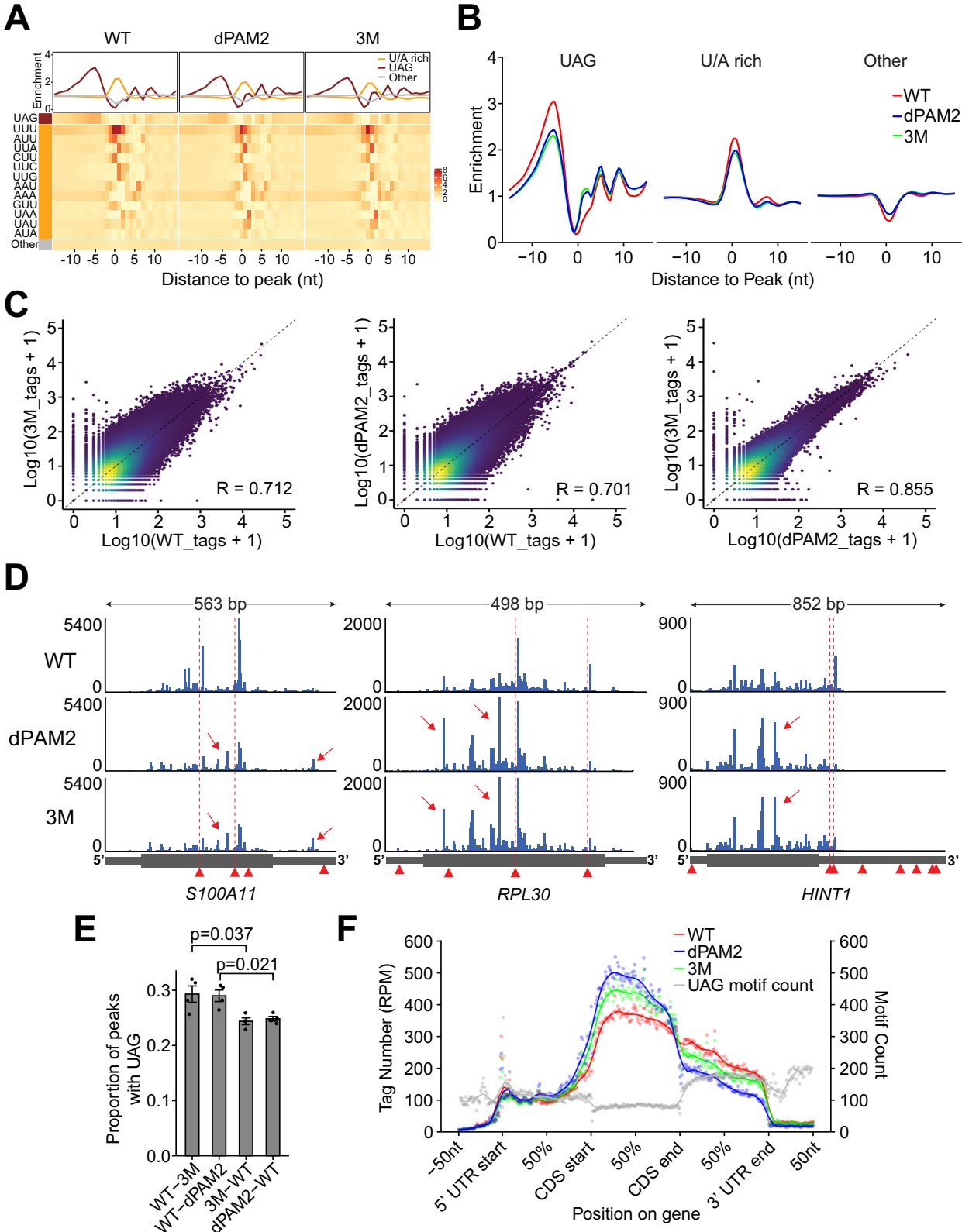

the deletion of PAM2 SLiM caused a profound reduction in UNK binding to 3′UTRs and increased targeting of the coding sequences (Fig. 5F). In contrast, the 3M mutation led to a more moderate but still significant upstream repositioning of UNK (Fig. 5F). This suggests that UNK position on mRNAs is controlled through association with effectors, with PABPC exerting a stronger influence compared to CCR4-NOT.

PABPC has a low nanomolar affinity for poly(A) RNA and is found largely at or very near the poly(A) tails[36,38]. However, factors such as PAIP2 or TNRC6 were reported to displace PABPC from mRNAs[39,40]. To determine whether UNK may function similarly, we first inspected the bulk interactions of PABPC1 with poly(A) tails (Supplementary Fig. 6H). UNK expression showed no effect on the pattern of ~27-nt footprints of PABPC on poly(A) tails in partially digested RNA co-precipitated

**Fig. 5 | Impact of effector interactions on RNA binding by Unkempt. A**, **B** Altered RNA sequence recognition by UNK mutants. **A** Heatmaps illustrate positional frequencies of the 64 possible trimers within UNK-binding sites between 15 nts upstream and downstream of the binding-site maxima for UNK$_{WT}$, UNK$_{dPAM2}$, and UNK$_{3M}$. Plots above the heatmaps profile the mean enrichment of different sets of trimers, considering the upstream UAG trimer (brown), the downstream U/A-rich trimers (orange), and all other trimers (gray; see "Methods" section). **B** Overlays of the mean enrichment profiles for each set of trimers shown in (**A**). **C** Density scatterplots comparing crosslink events per peak pairwise among the combined iCLIP replicates ($n = 4$) for UNK$_{WT}$, UNK$_{dPAM2}$, and UNK$_{3M}$. **D** Normalized UNK$_{WT}$, UNK$_{dPAM2}$, and UNK$_{3M}$ iCLIP coverage tracks for three UNK target transcripts showing mutant-specific reduced binding strength at UAG-containing binding sites and the emergence of UAG-less satellite peaks. Arrowheads indicate positions of all UAG motifs; positions of UAG motifs immediately upstream of UNK$_{WT}$ binding sites

are highlighted by dashed vertical lines. Arrows point to UAG-less satellite peaks. **E** Transcriptome-wide occurrence of the UAG trimer within 15 nts upstream of WT-specific (WT-3M and WT-dPAM2) or mutant-specific peaks (3M−WT and dPAM2-WT). Data are presented as mean ± SD. Statistical significance for the indicated comparisons considering four iCLIP replicates per condition was determined using Student's two-tailed *t*-test. **F** Proportional metatranscript analysis of iCLIP data showing the positional frequency of crosslink events for UNK$_{WT}$, UNK$_{dPAM2}$, and UNK$_{3M}$ on different segments of mRNA. Data points represent normalized crosslink events summarized over every percent of a given mRNA segment. Gray data points show total counts of the UAG trimer. Note the 5′-shift of RNA binding into coding sequences (CDS) by UNK$_{3M}$ and especially by UNK$_{dPAM2}$ despite the relative depletion of UAG in this mRNA segment. See "Methods" section for determination of statistical significance for pairwise comparisons in (**B**) and (**F**).

with PABPC1 following in vivo UV crosslinking (Supplementary Fig. 6H), indicating a generally intact binding of PABPC1 to poly(A) tails.

To obtain a sequence-specific view of PABPC1 binding, we carried out iCLIP of endogenous PABPC1 and focused on its unique binding sites known to cluster around polyadenylation signals[36]. As with the gross analysis of poly(A) tails (Supplementary Fig. 6H), we observed no overt changes in the binding pattern of PABPC1 upon expression of UNK (Supplementary Fig. 6I–K, Supplementary Data 4). However, UNK$_{3M}$ or UNK$_{dPAM2}$ showed weaker enrichment in the vicinity of PABPC1 binding sites than UNK$_{WT}$, consistent with the reduced presence of the UNK mutants on 3′UTRs (Fig. 5F, Supplementary Fig. 6K). These results support a model where PABPC strongly influences the distribution of UNK on mRNAs but not vice versa.

## The Unkempt–effector interface indirectly regulates steady-state mRNA levels

UNK is a translational repressor that has little effect on transcript stability[16]. As CCR4-NOT and PABPC are principal factors affecting mRNA translation and stability, we asked whether the interactions of these effectors with UNK mediate its regulatory input.

We first determined the imprint of UNK$_{WT}$, UNK$_{dPAM2}$, and UNK$_{3M}$ on the cellular transcriptome 24 h post-induction of expression (Supplementary Data 5). UNK$_{WT}$ perturbed steady-state mRNA levels of many transcripts with some bias toward downregulation (Fig. 6A). A correlative analysis of iCLIP data indicated relatively weak binding of the highly regulated transcripts by UNK$_{WT}$ and stronger targeting of transcripts that showed little regulation, again with a moderate preference for downregulated messages (Fig. 6A). Similar trends were also noted upon induction of UNK mutants, however, with UNK$_{dPAM2}$ affecting only about 60% and UNK$_{3M}$ less than 10% of the number of transcripts regulated by UNK$_{WT}$ (Fig. 6B–D). Thus, the largely indirect effect of UNK on steady-state mRNA levels relies heavily on its interactions with the CCR4-NOT complex and less on PABPC.

We then asked if UNK influences the shortening of mRNA poly(A) tails, a process known as deadenylation and in which PABPC and CCR4-NOT both play principal roles[24,25]. Using direct RNA sequencing, we derived mRNA poly(A) tail length estimates in different conditions of UNK expression in cells. Irrespective of UNK$_{WT}$, UNK$_{dPAM2}$, or UNK$_{3M}$ expression, we observed a length distribution consistent with relatively short tails of highly expressed mRNAs, a conserved feature of eukaryotic cells (Supplementary Fig. 7A, B, Supplementary Data 6)[41–43]. We found no correlation between the strength of mRNA targeting by UNK and mRNA poly(A) tail length, regardless of whether UNK$_{WT}$ or its mutants were expressed (Supplementary Fig. 7C). We conclude that UNK does not substantially impact the metabolism of mRNA poly(A) tails in cells and that the effects on steady-state mRNA levels are mediated indirectly by its effector interface (Fig. 6A–D, Supplementary Fig. 7D).

## SLiM-mediated contacts with CCR4-NOT are critical conduits of translational control

To test whether the interactions with CCR4-NOT and PABPC mediate the translational regulation by UNK, we conducted ribosome profiling experiments to evaluate the impact of the effector interface on translational efficiencies of mRNAs while also considering the strength of mRNA targeting by UNK. The expression of UNK$_{WT}$ resulted in a striking reduction in ribosome occupancy for the large majority of all significantly regulated mRNAs (91.6% or 2350 mRNAs; Fig. 6E, Supplementary Data 7). Notably, most of these transcripts were highly bound by UNK, whereas the few with gains in ribosome occupancy were not (Fig. 6E, Supplementary Fig. 7E), pointing to a strong and direct repressive effect of UNK on translation. A separate analysis that only considered transcripts with no changes in expression indicated comparable, if not greater, bias toward translational silencing (Supplementary Fig. 8A).

Similar analyses for UNK$_{dPAM2}$ and UNK$_{3M}$ revealed that CCR4-NOT and, to a lesser extent, PABPC are critical mediators of UNK-driven translational control (Fig. 6E–H, Supplementary Fig. 8B–G). Specifically, removing the interaction with PABPC reduced the number of significantly repressed transcripts by about 20% (Fig. 6F, H, Supplementary Fig. 8C, F), whereas disrupting binding to CCR4-NOT essentially eliminated UNK-mediated translational repression (Fig. 6G, H, Supplementary Fig. 8D, G). Markedly, we could pinpoint the critical interface on the CCR4-NOT effector as the removal of the CNOT9 subunit, which weakened the interaction with UNK (Supplementary Fig. 4B), rendered UNK$_{WT}$ incompetent for target repression (Supplementary Fig. 8H–M). Thus, the UNK–CCR4-NOT nexus is a critical conduit of translational repression for a large fraction of the cellular mRNA pool.

We also considered that the ability of the UNK–PABPC nexus to repress translation may be limited by the distance from the PABPC-binding sites on poly(A) tails and 3′UTRs. To test this, we considered the lengths of UNK targets whose translational silencing depends on UNK maintaining contacts with either PABPC or CCR4-NOT (Fig. 6E–G). Transcripts silenced by UNK$_{WT}$ but not by UNK$_{dPAM2}$ were significantly shorter than those with sustained repression ($p = 8.6e^{-05}$), whereas the large population of mRNAs derepressed due to weakened contacts between UNK$_{3M}$ and CCR4-NOT showed no such bias (Fig. 6I). These findings indicate differential functional requirements for UNK–effector interactions in the context of a translationally repressive RNP.

It is of interest to note that the direct effect of the studied perturbations on translation is closely matched by changes in the mRNA levels as well as the morphogenetic potential of UNK (Fig. 6A–H, Supplementary Figs. 3E, 8B–M). The principal implication is that much of UNK's cellular activity is coupled to its SLiM-mediated regulation of protein translation.

In conclusion, the identified critical disordered segments of UNK (Fig. 1) and their conserved interactions with its identified effectors

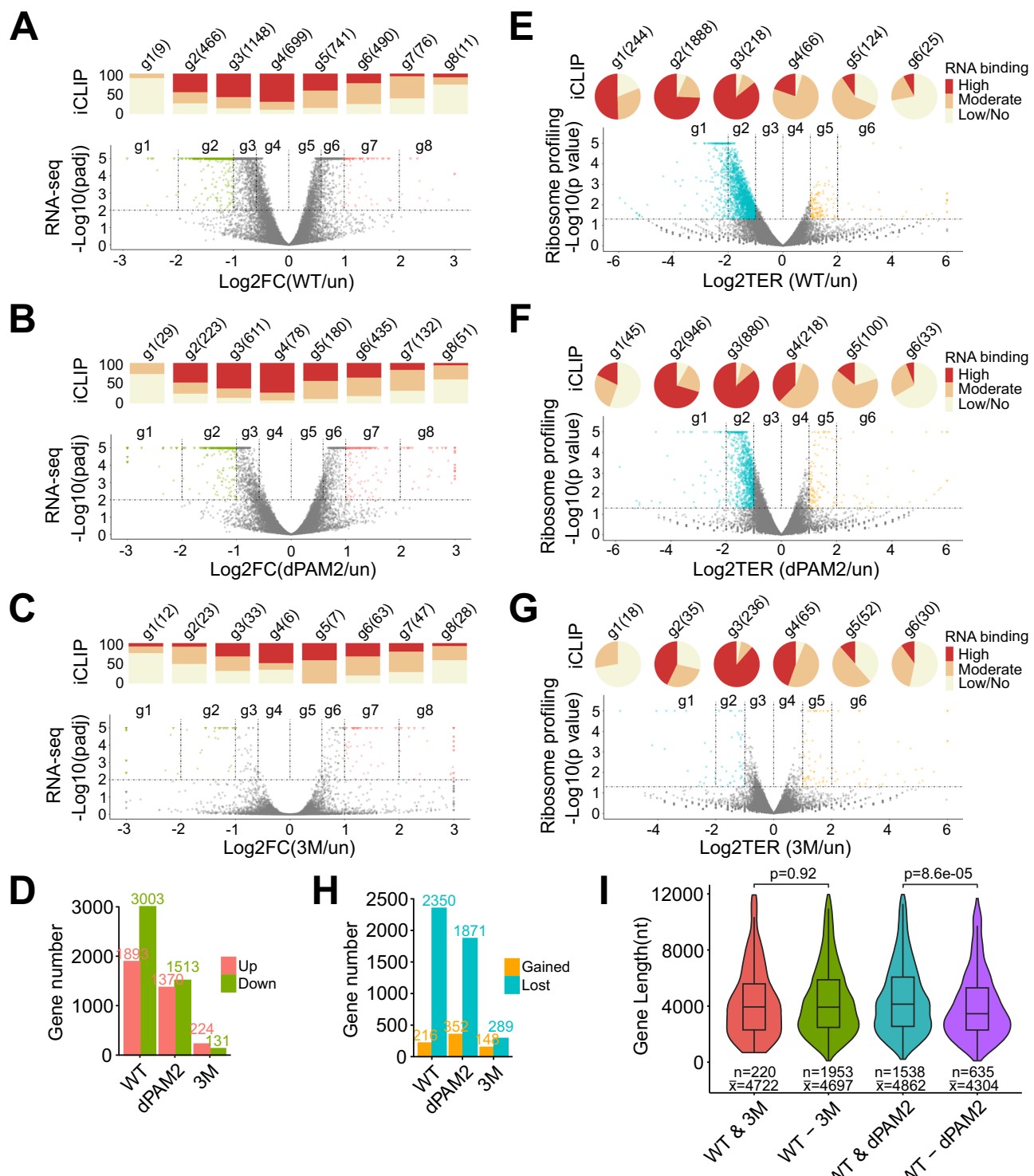

(Figs. 2–4) show a clear relationship with the RNA-binding capacity and regulation (Figs. 5, 6), providing a framework for future studies of the functional principles of RBPs.

## Discussion

How do RBPs interact with their effectors to instruct different types of RNA processing? To address this question, we introduce an integrative approach combining in vitro and in vivo biochemistry, structure prediction, and multiple levels of systems analyses to study the RBP–effector interface, taking as a model an essential RBP with distinct general molecular and cellular activities, Unkempt. This strategy

allowed us to identify critical effectors of UNK, define the interactions constituting the interface, and evaluate the functional contribution of these interactions to UNK's activity.

UNK forms multivalent interactions with its effectors, the CCR4-NOT complex and PABPC. This includes an extensive IDR that contains three effector-binding SLiMs, two of which, SLiM 1 and SLiM 2, specifically interact with CCR4-NOT, and one, a PAM2-like SLiM, that interacts with PABPC, as well as a C-terminal coiled-coil domain that induces UNK to homodimerize. Each of the CCR4-NOT–binding SLiMs can contact the NOT9 and the NOT module of the CCR4-NOT complex. This suggests an assembly of a functionally competent RNP that is held

**Fig. 6 | Post-transcriptional regulation at the Unkempt–effector interface.**
**A**–**D** Impact of UNK–effector interactions on steady-state mRNA levels.
**A**–**C** Volcano plots showing differential mRNA abundances between uninduced cells (un) and cells expressing UNK$_{WT}$ (**A**), UNK$_{dPAM2}$ (**B**), or UNK$_{3M}$ (**C**) (RNA-seq data; $n = 3$). Significantly regulated transcripts (p adj. <0.01) are binned into eight groups (g1-g8) according to the strength and sense of their regulation, with transcripts showing twofold or larger changes in abundance highlighted in color. Bar charts above the volcano plots indicate proportions of mRNAs bound highly, moderately, or lowly/not bound in each transcript group (iCLIP data). The total numbers of transcripts in each group are indicated. FC, fold change. **D** Total numbers of up- or downregulated transcripts for each comparison shown in **A**–**C** (p adj. <0.01). **E**–**H** Direct translational repression mediated by UNK–effector interactions. **E**–**G** Volcano plots summarize ribosome profiling analyses showing differential ribosome occupancies of transcripts between uninduced cells and cells expressing UNK$_{WT}$ (**E**), UNK$_{dPAM2}$ (**F**), or UNK$_{3M}$ (**G**) ($n = 2$). Significantly regulated

transcripts ($p < 0.05$) are binned into six groups (g1–g6) according to the strength and sense of their regulation, with transcripts showing twofold or larger changes in ribosome occupancy highlighted in color. Pie charts indicate RNA-binding information (iCLIP data) and transcript numbers, as in (**A**–**C**). TER, translational efficiency ratio. **H** Total numbers of transcripts with gained or lost ribosome occupancy for each comparison shown in (**E**–**G**) ($p < 0.05$). **I** Loss of contact with PABPC leads to preferential translational derepression of shorter transcripts. Violin plots show the distribution of mRNA lengths in groups of UNK-targeted transcripts that are translationally significantly repressed ($p < 0.05$) by UNK$_{WT}$ and UNK$_{3M}$ (WT & 3M) or by UNK$_{WT}$ and UNK$_{dPAM2}$ (WT & dPAM2), and those that are uniquely repressed by UNK$_{WT}$ but not UNK$_{3M}$ (WT-3M) or by UNK$_{WT}$ but not UNK$_{dPAM2}$ (WT-dPAM2). The numbers of transcripts in each group (n) and their average length (x) are indicated. Statistical significance was calculated using Student's two-tailed t-test. See "Methods" section for significance calculations in (**A**–**C**) and (**E**–**G**) and definition of box plots in (**I**).

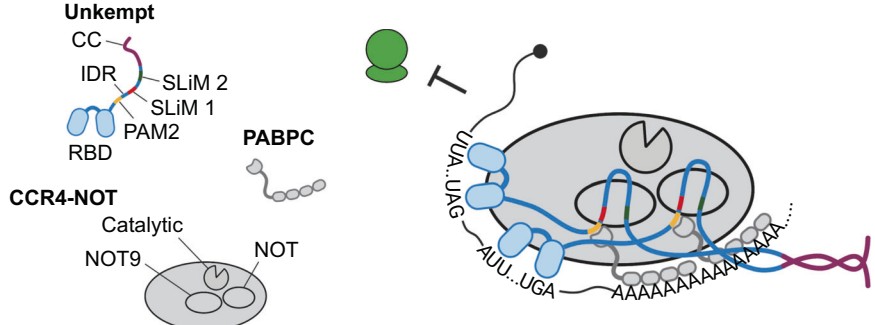

**Fig. 7 | Model for regulation at the UNK–effector interface.** Multivalent, high-avidity interactions between UNK and its effectors, CCR4-NOT and PABPC, are enabled by IDR-embedded SLiMs and the C-terminal coiled-coil (CC) domain of UNK. See also Supplementary Fig. 9. Created with BioRender.com.

together by multivalent, SLiM-mediated RBP–effector interactions (Fig. 7, Supplementary Fig. 9A–C). In the modeled RNP, two RNA-binding domains are brought together and can bind to the same or different molecules of mRNA, and this multivalency may facilitate the assembly of a larger RNP (Supplementary Fig. 7C).

We identify CCR4-NOT as the principal effector of UNK, with PABPC in a supportive regulatory role. Disrupting the UNK–CCR4-NOT interface essentially eliminates the signature activity of UNK at multiple scales in cells, including its effects on protein translation, perturbation of the transcriptome, and cell morphogenesis. Converting UNK to a monomeric state results in a complete loss of its morphogenetic activity. Although interrupting the UNK–PABPC contacts results in a weaker reduction of UNK's activity, the effect is again consistent across scales and thus in line with the central regulatory role of the UNK–effector interface.

For many RBPs, low proportions of observed versus expected RNA-binding events suggest that additional determinants of RNA binding must exist[33,44,45]. Effector interactions have traditionally been viewed as serving a recruiting role and have not been thought to feed back onto RNA binding by the recruiting RBP[2]. Unexpectedly, we find that interactions with CCR4-NOT or PABPC exert a substantial, two-prong auxiliary effect on RNA binding by UNK. First, both effectors assist UNK with the specificity of RNA sequence recognition; this is seen globally with compromised effector contacts leading to reduced binding of the critical UAG and the adjacent U/A-rich motifs, as well as locally by the emergence of numerous UAG-less satellite peaks (Fig. 5A–E). The remarkably similar defects in RNA binding by UNK$_{dPAM2}$ or UNK$_{3M}$ suggest that PABPC and CCR4-NOT may both stabilize UNK on mRNA (Supplementary Fig. 9B).

Independent of RNA sequence recognition, PABPC and, to a lesser extent, CCR4-NOT appear to control the distribution of UNK on mRNAs by facilitating its binding to 3′UTRs on targets. This is

supported by a transcriptome-wide repositioning of UNK mutants to coding sequences (Fig. 5F), which may be assisted by UNK's ability to bind endogenously paused ribosomes[16]; this could also explain the relative paucity of UNK in 5′UTRs of its targets. Along with the largely unaltered RNA binding by PABPC, the observed repositioning suggests that the 3′-anchored PABPC secures UNK to 3′UTRs rather than it being recruited by UNK, similar to how PABPC promotes the association of miRISC with mRNAs or the positioning of Makorin 1 RBP upstream of premature poly(A) tails[14,46]. In contrast to PABPC, the more subtle 5′-shift of UNK$_{3M}$ and suppression of mRNA targets of all lengths (versus the regulatory bias of PABPC toward shorter mRNAs; Fig. 6I) point to recruitment of CCR4-NOT to UNK-binding sites on mRNAs. We also note that UNK$_{dPAM2}$ or UNK$_{3M}$ target similar but not identical sets of transcripts compared to UNK$_{WT}$ (Supplementary Fig. 6G), implying that effector contribution to RNA binding may itself play a regulatory role.

The subunits of CCR4-NOT with which UNK interacts via SLiMs are also known to be targeted by divergent SLiMs of other RBPs and non-RBPs, indicating an independent but convergent evolution of IDR-embedded SLiMs of UNK[2,9,47–52]. However, unlike the multivalent interface between the UNK homodimer and CCR4-NOT described here, only one or at most two contacts have been experimentally validated between any other protein and CCR4-NOT, although more numerous contacts are suspected to exist[9,37]. Given the generally weak affinity of SLiM-mediated interactions, the large surface of the CCR4-NOT complex, and the commonly reported recruitment of CCR4-NOT by RBPs, it is plausible that additional, yet unidentified contacts with CCR4-NOT facilitate regulation by RBPs.

The existence of multi-purpose SLiMs suggests an economical evolutionary adaptation serving to eliminate a need to maintain a separate SLiM for each effector contact. This would permit combinatorial flexibility of RBP–effector interactions or facilitate their

synchronous regulation, e.g., via post-translational modifications (Supplementary Fig. 9A). Remarkably, structural predictions based on evolutionary data suggest that the key protein features participating in the formation of UNK–CCR4-NOT contacts already existed in the earliest known UNK ortholog that emerged more than 500 million years ago and was specific for the UAG motif with some level of cell-morphogenetic activity (Supplementary Fig. 5E, F)[17,53].

Past studies of RBP- or miRNA-mediated gene silencing commonly relied on tethering assays and reporter transcripts to investigate the translational repression by the CCR4-NOT complex that is decoupled from its impact on mRNA stability[9,54–57]. We comprehensively show on a transcriptome-wide scale that mRNA regulation via CCR4-NOT in cells is not necessarily accompanied by deadenylation and mRNA decay. In the case of UNK, we speculate that deadenylation may be inhibited by the extensive interactions of IDR with the NOT9 and NOT modules, which are known to directly stimulate deadenylation by CCR4-NOT[28,58].

Several aspects of our study merit further consideration. For instance, we find that effector interactions are required for the accuracy of RNA sequence recognition by UNK, but cannot explain why this is so, why both PABPC and CCR4-NOT show a similar requirement, and why some but not other UNK-binding sites require effector contacts. Rationalizing these observations will provide new fundamental knowledge about determinants of RNA-binding-site selection and functional organization of RNPs. Toward this goal, it will be interesting to determine the in vivo positions of CCR4-NOT on mRNAs and its repositioning by UNK, evaluate whether or not both effectors associate with UNK at its mRNA-binding sites, and carry out structural studies of UNK RNPs, especially those reconstituted in vitro from purified components.

The striking reliance on multivalent interactions between UNK and the multisubunit CCR4-NOT complex, multiplied by a high number of RNA-binding sites that generally track with efficient translational repression by UNK, could conceivably lead to molecular-scale condensation. It can be envisaged that endogenous UNK, in the context of such condensation, competes with the deadenylation machinery composed of CCR4-NOT and PABPC[24], possibly during embryonic development of the brain.

One might speculate that the reporter plasmid-linked transcriptional activity of the principally cytoplasmic UNK would at least in part rely on its recruitment of CCR4-NOT, which is known to participate in diverse gene regulatory processes, including control of gene transcription[32,59]. Curiously, however, we found that the Gal4-UNK fusion is comparably transcriptionally active in the CNOT9 knockout as it is in wild-type cells, suggesting that the reporter-linked transcriptional activity of UNK, unlike its translational role, does not rely on the CNOT9 subunit. It is plausible that, in the absence of CNOT9, the CCR4-NOT complex is efficiently recruited to the promoter-tethered UNK via its NOT module to drive transcription of the reporter gene. Alternatively, transcriptional effectors other than CCR4-NOT might interact with the critical residues of UNK in the chromatin environment.

Although we were not able to find any evidence for transcriptional activity of native UNK, we leave open a possibility that the regulatory repertoire of UNK encompasses transcription. If this is the case, then how might an RBP combine potent transcriptional activation with translational repression? Considering the results of our systems analyses and the morphogenetic activity of UNK, we envision two possibilities. The first is that UNK could be transcriptionally inducing only a small subset of genes but translationally repressing a much broader cohort of (different) genes. Alternatively, UNK could conceivably activate transcription of a particular protein-coding gene and stay bound to the resulting transcript while keeping it translationally silent until release for localized translation, for instance, at the growth cones of polarizing cells. It remains to be determined whether any of these mechanisms is operational in cells. While beyond the scope of this study, links between the molecular and cellular biology of UNK, including its control of cell morphogenesis, present intriguing avenues to explore in future studies.

## Methods

### Doxycycline-inducible cell lines

Human cell lines, including SH-SY5Y (CRL-2266, ATCC), HeLa (CCL-2, ATCC), and 293T (CRL-3216, ATCC), were maintained in DMEM medium supplemented with 10% FBS and penicillin/streptomycin at 37°C and 5% $CO_2$. Cells were authenticated by ATCC using STR profiling and were regularly tested for mycoplasma contamination during experimentation.

For evaluation of the morphogenetic activity of UNK, Dox-inducible HeLa cells were created via infection with an all-in-one lentivirus expressing a puromycin resistance gene (PuroR), advanced reverse tetracycline-controlled transactivator protein (rtTA$^{Ad}$), and a TREtight-driven transcript encoding GFP alone or GFP and either Flag-HA-tagged UNK$_{WT}$ or UNK mutants (pLIX-IRES-GFP; see Fig. 1A and Plasmid constructs). Dox-inducible HeLa cells used in all other experiments were generated analogously, using a similar all-in-one lentivirus that did not express GFP (pLIX-403; Addgene_41395), and were made monoclonal via single-cell sorting to ensure comparable inducible expression of UNK in cells within a population and among populations expressing Flag-HA-tagged UNK$_{WT}$, UNK$_{dPAM2}$, or UNK$_{3M}$. To induce transgene expression, puromycin-resistant cells were treated with doxycycline (Millipore Sigma) at 1 μg/ml. CRISPR/Cas9-mediated knockout of CNOT9 was achieved via transduction with a lentivirus for expression of gRNA, Cas9, and a blasticidin resistance gene (lentiCRISPR v2-Blast; Addgene_83480). Successful CRISPR/Cas9-mediated genome editing was monitored in single-cell clones by PCR and sequencing of the genomic locus. All lentiviral particles were produced in 293T cells by co-transfection of a lentiviral expression vector, the lentiviral packaging vector pCMV delta R8.2 (Addgene_12263) and the pMD2.G vector (Addgene_12259) with polyethylenimine (Polysciences, 23966-100; pH7.0). Growth medium was exchanged 16 h post-transfection. After 2 days, virus-containing supernatant was filtered through a 0.45 μm syringe filter and used for transduction.

### Plasmid constructs

Plasmids for transient expression of Gal4-tagged UNK mutants (Supplementary Figs. 2, 3) were created using the Gateway cloning strategy where UNK mutants in the pENTR/D-TOPO vector backbone (Thermo Fisher Scientific, K240020) were transferred in an LR reaction using the Gateway LR Clonase II Enzyme mix (Thermo Fisher Scientific, 11791020) into the pDEST-pcDNA3-Gal4 vector (generated by insertion of the ccdB cassette into the pcDNA3-Gal4 construct[60]; gift from Fei Lan) following manufacturer's protocol. The entry clones were generated by first inserting the full-length mouse UNK$_{WT}$ that was amplified by PCR from the pTtight-UNK-IGPP vector[16] using the Gateway BP Clonase II Enzyme Mix (Thermo Fisher Scientific, 11789020) into the pENTR/D-TOPO vector, yielding pENTR-UNK. Full-length UNK deletion mutants and residue mutants F504A, L522A, 3M, WFAA, and W622A (Supplementary Data 1) were prepared by mutating the pENTR-UNK vector. Specifically, the deletion mutants were generated by PCR with oligos flanking the deleted regions and amplification of the entire plasmid. The resulting reactions were treated with DpnI and transformed into One Shot TOP10 *E. coli* (Thermo Fisher Scientific, C404003). The above residue mutants were created by site-directed mutagenesis following instructions provided in the QuickChange XL Site-Directed Mutagenesis Kit manual (Agilent Technologies). All other full-length UNK residue mutants, including 11DE-A, 10FY-A, 7KH-A, and E8 (Supplementary Data 1), were created by replacing IDR$_{WT}$ in pENTR-UNK with corresponding mutant IDRs synthesized as GeneArt Strings DNA fragments (Thermo Fisher Scientific). To enable the

replacements, BamHI and BspEI restriction enzyme cut sites were introduced by silently mutating UNK sequences immediately 5′ and 3′ to the IDR, respectively, by site-directed mutagenesis. The replacements were performed by cutting the resulting entry vector with BamHI and BspEI to release IDR$_{WT}$ and clone in either of the mutant IDRs amplified by PCR from the synthetic DNA fragments. All full-length UNK$_{WT}$ and IDR$_{WT}$ truncation mutants analyzed in Supplementary Figs. 2 and 3 (see also Supplementary Data 1), as well as IDR residue mutants F504A, L522A, 3M, 11DE-A, 10FY-A, and 7KH-A were created by PCR amplification of the corresponding UNK segments from the above entry vectors followed by their insertion into the EcoRI- and XbaI-cut pENTR vector. IDR residue mutants D520S and 3D-S were created by site-directed mutagenesis of pENTR-IDR. IDR residue mutants 5DE-S, 7DE-S, 41S-A, 23LI-A, and 17P-A were ordered as synthetic DNA fragments and cloned into pENTR as above.

For the expression of MBP-IDR-Strep proteins in bacteria (Fig. 3), WT or mutant IDR were amplified from the above entry vectors or a synthetic DNA fragment encoding IDR with V511E/I515E/L522E substitutions such that two StrepII tags (GSGWSHPQFEKGSWSHPQFEK) were added in-frame straight after the C-terminal residue of IDR in each protein. The amplicons were then inserted individually in the pnYC-NvHM_M plasmid (Addgene_146932) between NdeI and MfeI sites. The same strategy was employed for cloning of UNK$_C$, UNK$_C$ E6, and UNK$_C$ E8 (Fig. 4), which were amplified from pENTR-UNK (for UNK$_C$) or synthetic DNA fragments (for UNK$_C$ E6 and UNK$_C$ E8) for insertion in the pnYC-NvHM_M plasmid.

Plasmids for the expression in insect cells of the full-length UNK fused C-terminally to two StrepII tags (Figs. 3, 4, Supplementary Fig. 5) were generated by insertion of the full-length UNK amplified from the pENTR-UNK vector by PCR, which also introduced two C-terminal StrepII tags, in the pLIB plasmid (Addgene_80610) between BamHI and SalI sites.

All-in-one lentiviral plasmids for Dox-inducible expression of UNK without GFP in HeLa cells were created by insertion of the Flag-HA-tagged full-length WT or mutant UNK amplified from the corresponding entry vector into the pLIX_403 plasmid (Addgene_41395) between NheI and AgeI sites. For Dox-inducible expression of GFP with or without UNK, the pLIX-IRES-GFP plasmid was first created by subcloning the IRES-GFP segment from pTRE-tight-IRES-GFP-PGK-Puro[16] into pLIX_403 between NheI and AgeI sites. WT or mutant Flag-HA-tagged UNK, amplified from the entry vectors above, were then cloned individually in pLIX-IRES-GFP between NheI and MluI sites.

To generate a plasmid for stable knockdown of the endogenous UNK in SH-SY5Y cells (Supplementary Fig. 1), an shRNA targeting human *UNK* gene (targeted sequence: CAGGTACCACCTTCGTTACTA) was cloned in the pLKO.1 puro plasmid (Addgene_8453) between AgeI and EcoRI sites[61]. We used the scramble shRNA plasmid (Addgene_1864) for the expression of non-targeting control shRNA (Supplementary Fig. 1).

The human *CNOT9*-targeting or non-targeting control guide sequences were introduced into the BsmBI-digested lentiCRISPR v2-Blast plasmid (Addgene_83480) as pairs of annealed oligos[62]. gRNA-targeted sequence in CNOT9 was CCCATGCTGTGGCATTCATT.

### Transfection of siRNAs
HeLa cells inducibly expressing GFP and Flag-HA-tagged UNK$_{WT}$ were seeded in 6-well dishes and transfected 24 h later at about 40% confluence using the *Trans*IT-X2 Dynamic Delivery System (Mirus, MIR6003) with a pool of siRNAs targeting CNOT7 (Horizon, L-012897-00-0005), CNOT8 (Horizon, L-018791-00-0005), CNOT9 (Horizon, L-019972-00-0005), or a non-targeting siRNA pool (Horizon, D-001206-13) at 50 nM. Cells were induced with Dox at 24 h after transfection and cell morphologies or the efficiency of knockdown were evaluated at 48 h after induction.

### Dual-luciferase reporter assay
Dual-luciferase reporter assays were performed by co-transfecting 400 ng of a Gal4-tagged UNK mutant-expressing plasmid, 200 ng of the pGL4.35[luc2P/9XGAL4UAS/Hygro] Vector (Promega, E1370), and 20 ng of the control pRL-TK Vector (Promega, E2241) into 293T cells using Lipofectamine 2000 Transfection Reagent (Themo Fisher Scientific, 11668019). Forty-eight h after transfection, cells were harvested and processed using Dual-Luciferase Reporter Assay System (Promega, E1960) according to the manufacturer's instructions. The emitted luminescence was detected using SpectraMax L Luminescence Microplate Reader (Molecular Devices).

### RT-qPCR analysis
Total RNA was extracted from samples equivalent to those used for the dual-luciferase assays using TRIzol Reagent (Themo Fisher Scientific, 15596018) and Direct-zol RNA Miniprep (Zymo Research, R2050) according to the manufacturer's instructions. cDNA was prepared from equal amounts of RNA using PrimeScript RT Reagent Kit (Takara, RR037A) following manufacturer's instructions. qPCR was performed using PowerUp Sybr Green Master Mix (Thermo Fisher Scientific, A25742) to amplify the cDNA on the CFX Connect Real-Time PCR Detection System at the annealing temperature of 63 °C. Relative firefly luciferase mRNA levels were normalized to relative expression levels of the *RPS18* gene that was used as an internal control. Primers used for firefly luciferase were FF-F: GAGCTATTCTTGCGCAGCTT and FF-R: CCTCACCTACCTCCTTGCTG; primers for *RPS18* were RPS18-F: GATGGGCGGCGGAAAATAG and RPS18-R: GCGTGGATTCTGCATAATGGT.

### Immunofluorescence
SH-SY5Y cells and HeLa cells ectopically expressing Flag-HA-tagged UNK$_{WT}$ or UNK$_{3M}$ were fixed in 4% paraformaldehyde for 10 min at room temperature, permeabilized in 0.5% Triton X-100, blocked in 5% goat serum, and probed with anti-UNK (Millipore Sigma, HPA023636) or anti-HA antibodies (multiple lots) at 1:250 dilution at 4 °C for 24 h. After an overnight incubation, the cells were probed with fluorochrome-conjugated secondary antibodies for 1 h at room temperature and mounted using VECTASHIELD Antifade Mounting Medium with DAPI (Vector Laboratories, H-1200-10). Images were taken with the LSM 880 confocal microscope (Zeiss).

### Quantification of cell morphologies
Cell morphologies were quantified essentially as reported previously[16]. Briefly, after 48 h of incubation with Dox, HeLa cells inducibly expression either GFP alone or GFP and WT or mutant UNK were imaged and the axes of GFP-positive cells were measured with Adobe Illustrator software (Adobe). The morphologies of at least 50 GFP-positive cells were quantified for each induced transgene by calculation of their axial ratios, y/x, where y is the length of the absolute longest cellular axis and x is the length of the longest axis perpendicular to the y axis.

### SDS-PAGE and western blot analysis
Whole-cell lysates and eluates from immunoprecipitations were run on 10% SDS-polyacrylamide gels and transferred to supported nitrocellulose membrane (Bio-Rad) by standard methods. Membranes were then blocked for 1 h in 5% non-fat dry milk in 1× TBS with 0.1% Tween-20 (TBST), rinsed, and incubated with primary antibody diluted in 3% BSA in TBST overnight at 4 °C. The following primary antibodies, all from multiple lots, except anti-CNOT7, were used: anti-Flag (Millipore Sigma, F1804), anti-UNK (Millipore Sigma, HPA023636), anti-CNOT1 (Proteintech, 14276-1-AP), anti-CNOT2 (Cell Signaling Technology, 34214), anti-CNOT3 (Proteintech, 11135-1-AP), anti-CNOT7 (gift of A.B. Shyu)[63], anti-CNOT9 (Fine Test, FNab07487), anti-PABPC (Abcam, ab21060), and anti-β-Actin-peroxidase (Millipore Sigma, A3854). All

primary antibodies were used at 1:1,000, except anti-β-Actin-peroxidase, which was used at 1:20,000. Blots were washed in TBST, incubated with HRP-conjugated secondary antibodies in 5% milk in TBST for 1 h (except for anti-β-Actin-peroxidase antibody), and washed again. HRP signal was detected by Western Lightning Plus chemiluminescent substrate (NEL103001EA).

## Co-immunoprecipitation from cell lysates

For co-IP experiments, HeLa cells inducibly expressing Flag-HA-tagged UNK were treated for 24 h with Dox. Uninduced samples were processed in parallel. Cells were harvested, washed once with PBS, and lysed in whole-cell lysis buffer (20 mM HEPES-KOH pH 7.9, 10% glycerol, 300 mM KCl, 0.1% IGEPAL, 1 mM DTT) supplemented with cOmplete Protease Inhibitor Cocktail (Millipore Sigma, 11697498001) for 30 min at 4 °C. Supernatants were cleared off debris by a 30-min centrifugation at $17,000 \times g$ at 4 °C. The lysates were then mixed with an equal volume of no-salt lysis buffer (20 mM HEPES-KOH pH 7.9, 10% glycerol, 0.1% IGEPAL, 1 mM DTT) supplemented with cOmplete Protease Inhibitor Cocktail to lower the final salt concentration to 150 mM KCl (IP buffer), added to anti-Flag or normal mouse IgG antibody-conjugated Protein G Dynabeads (Invitrogen 10003D), and rotated for 2 h at 4 °C. To prepare antibody-conjugated magnetic beads, 50 µl of Protein G Dynabeads per experiment were washed with the IP buffer, resuspended in 100 µl IP buffer with 2 µg antibody, rotated at room temperature for 45 min, and washed twice with the IP buffer before being mixed with the cleared lysate. After the IP, the beads were washed thoroughly with the IP buffer and the bound proteins were eluted with 200 µg/ml Flag (DYKDDDDK) peptide (GenScript, RP10586) in thermomixer at 4 °C, shaking at 1250 rpm for 1 h. The eluates were analyzed by western blotting.

## Protein complex purification

To purify protein complexes of UNK$_{WT}$, UNK$_{dPAM2}$, and UNK$_{3M}$, approximately 300 million HeLa cells per experiment were harvested at 24 h of induction with Dox, flash-frozen in liquid nitrogen, and stored at −80 °C until use. Cells were resuspended in buffer A (20 mM HEPES-KOH pH 7.9, 10% glycerol, 300 mM KCl, 0.1% IGEPAL, 1 mM DTT, cOmplete Protease Inhibitor Cocktail; 100 µl of buffer A was used per $10^6$ cells) and rotated at 4 °C for 30 min. The lysates were centrifuged at $17,000 \times g$ for 30 min at 4 °C, supernatants were collected, and dialyzed in dialysis buffer (20 mM Tris-Cl, pH 7.3, 100 mM KCl, 0.2 mM EDTA, 20% Glycerol, 0.2 mM PMSF, 10 mM beta-mercaptoethanol) for 1 h at 4 °C. The lysates were centrifuged at $17,000 \times g$ for 30 min at 4 °C, then 250 µl anti-Flag M2 affinity gel (Millipore Sigma, A2220) was added and the mixture was rotated for 2 h at 4 °C. The affinity gel was then washed with TAP-wash buffer (50 mM Tris-Cl, 100 mM KCl, 5 mM MgCl$_2$, 10% Glycerol, 0.2 mM PMSF, 0.1% NP40). The bound proteins were eluted with Flag peptide (200 µg/ml; GenScript, RP10586) in thermomixer at 4 °C, shaking at 1250 rpm for 1 h. The eluate was mixed with anti-HA magnetic beads (Thermo Fisher Scientific, 88837) and rotated for 2 h at 4 °C. The beads were washed with TAP-wash buffer and proteins were eluted using HA peptide (200 µg/ml; GenScript, RP11735) by shaking the beads in thermomixer at 1400 rpm for 45 min at 30 °C. The eluate was TCA-precipitated and analyzed by mass spectrometry.

## Mass spectrometry

Fifty µl of 50 mM ammonium bicarbonate with 10% acetonitrile were added to the dry tubes containing the TCA-precipitated protein and gently vortexed. Next, 10 µl (20 ng/µl) of modified sequencing-grade trypsin (Promega, V5111) was spiked into the solutions and the samples were incubated at 37 °C overnight. Samples were acidified by spiking in 5 µl 20% formic acid solution and then desalted by a STAGE tip[64]. On the day of analysis, the samples were reconstituted in 10 µl of HPLC solvent A. A nano-scale reverse-phase HPLC capillary column was created by packing 2.6 µm C18 spherical silica beads into a fused silica capillary (100 µm inner diameter, 30 cm in length) with a flame-drawn tip[65]. After equilibrating the column, each sample was loaded via a Famos auto sampler (LC Packings, San Francisco CA) onto the column. A gradient was formed and peptides were eluted with increasing concentrations of solvent B (97.5% acetonitrile, 0.1% formic acid). As peptides eluted, they were subjected to electrospray ionization and then entered into an LTQ Orbitrap Velos Elite ion-trap mass spectrometer (Thermo Fisher Scientific). Peptides were detected, isolated, and fragmented to produce a tandem mass spectrum of specific fragment ions for each peptide. Peptide sequences and hence protein identities were determined by matching protein databases with the acquired fragmentation pattern by the software program Sequest (Thermo Fisher Scientific)[66]. All databases included a reversed version of all the sequences and the data was filtered at between 1% and 2% peptide false discovery rate.

## Recombinant protein expression and purification

Full-length mouse UNK with two C-terminal StrepII tags was produced in Sf21 insect cells using the MultiBac baculovirus expression system as previously described (Plasmid constructs)[67,68]. In brief, DH10-EmBacY cells were transformed with pLIB-UNK, transposition onto the baculoviral genome was selected by blue-white screening, the bacmid DNA was isolated and transfected into Sf21 cells to generate baculovirus. Sf21 cells were grown to a density of $2 \times 10^6$ cells/ml at 27 °C in Sf900II medium (Thermo Fisher Scientific), infected with the V1 UNK stock of baculovirus, and harvested 48 h after they stopped dividing. Cells were resuspended in lysis buffer (50 mM HEPES, 500 mM NaCl, pH 7.5) and lysed using a Branson Ultrasonics Sonifier SFX550. The lysate was cleared by centrifugation at $40,000 \times g$ for 1 h at 4 °C and filtered through 0.45 µm syringe-driven filters (Millipore). The cleared and filtered lysate was loaded onto a 1 ml StrepTrap XT column (Cytiva). The bound protein was eluted in one step with binding buffer (50 mM HEPES, 200 mM NaCl, pH 7.5) supplemented with 50 mM biotin.

An MBP-tagged C-terminal fragment of UNK (residues 637–810) was produced in *E. coli* BL21 (DE3) Star cells (Thermo Fisher Scientific) in LB medium at 20 °C as a fusion protein carrying an N-terminal His6-MBP tag and two C-terminal StrepII tags (Plasmid constructs). Cells were resuspended in lysis buffer (50 mM HEPES, 500 mM NaCl, 30 mM Imidazole, pH 7.5) and lysed using a Branson Ultrasonics Sonifier SFX550. The lysate was cleared by centrifugation at $40,000 \times g$ for 1 h at 4 °C. The cleared lysate was loaded onto a 5 ml HisTrap column (Cytiva). The bound protein was eluted over a linear gradient with elution buffer (50 mM HEPES, 200 mM NaCl, 500 mM Imidazole, pH 7.5). The final step was size exclusion chromatography on a Superdex 200 26/600 column in a buffer containing 10 mM HEPES, 200 mM NaCl, 2 mM DTT, pH 7.5. In addition, two mutated constructs UNK$_{E6}$ (residues 637–810 with L650E, L654E, I661E, A682E, L706E, L713E substitutions) and UNK$_{E8}$ (residues 637–810 with L650E, L654E, I661E, A682E, L706E, L713E, L740E, L754E substitutions) were produced and purified in the same manner as the WT version.

To prepare the recombinant, thermostable 5′-deadenylase (Hnt3p protein from a thermophilic eukaryote *K. marxianus*) used in ribosome profiling experiments, BL21 (DE3) bacteria were transformed with the pNTK576-pET28a-His6x-KmHnt3 plasmid (gift from Nicholas Ingolia)[69]. Individual colonies were picked and 500 ml cultures were grown to an OD of 0.4. Liquid cultures were then induced with 500 µl of 1 M IPTG and transferred to a shaker at 16 °C for 18 h. Cultures were pelleted by centrifugation at $3,000 \times g$ for 20 min at 4 °C and the pellets were flash-frozen with liquid nitrogen. To lyse the cells, pellets were maintained on ice and resuspended in 15 ml lysis buffer (500 mM NaCl, 0.5% NP40 (Igepal), 10 mM Imidazole, 20 mM HEPES, 10 mM bMe, pH 7.5). Resuspended pellets were sonicated for a total of 90 s then centrifuged at $10,000 \times g$ for 20 min at 4 °C. The supernatant was collected and incubated with 1.5 ml of lysis buffer-equilibrated Ni-NTA

beads for batch binding. The slurry was rotated for 1 h at 4 °C and then spun at 1096 × g for 3 min. The supernatant was carefully removed and the Ni-NTA beads were washed with 10 ml high-salt wash buffer (1 M NaCl, 20 mM Imidazole, 10 mM bMe, 20 mM HEPES, pH 7.5) followed by low-salt wash buffer (10 mM NaCl, 20 mM Imidazole, 10 mM bMe, 20 mM HEPES, pH 7.5). Beads were incubated for 5 min with 1.5 ml elution buffer (10 mM NaCl, 250 mM Imidazole, 10 mM bMe, 20 mM HEPES, pH 7.5) and the elution fractions were collected. This process was repeated a total of three times. Glycerol was added to each collected fraction to a final concentration of 10% before flash-freezing in liquid nitrogen for long-term storage.

## StrepTactin pull-down assay

StrepII-tagged MBP, as well as StrepII-tagged and MBP-tagged UNK IDR (residues 467–640) WT and mutant constructs were produced in *E. coli* BL21 (DE3) Star cells (Thermo Fisher Scientific) grown in auto-induction medium overnight at 37 °C. Cells were resuspended in lysis buffer (50 mM HEPES, 500 mM NaCl, pH 7.5) and lysed using a Branson Ultrasonics Sonifier SFX550, the lysate was then cleared by centrifugation at 40,000 × g for 1 h at 4 °C. The cleared lysate or purified UNK was incubated with StrepTactin Sepharose resin (Cytiva, 28935599). After a 1-h incubation beads were washed twice with 50 mM HEPES, 500 mM NaCl, pH 7.5, 0.03% Tween, once with 50 mM HEPES, 500 mM NaCl, pH 7.5, and once with binding buffer (50 mM HEPES, 200 mM NaCl, pH 7.5). Purified modules of the human CCR4-NOT complex, prepared as previously described[28], or purified PABPC1 were added to the bead-bound proteins. After a 1-h incubation, beads were washed four times with binding buffer and proteins were eluted with 50 mM biotin in binding buffer. The eluted proteins were analyzed by SDS-polyacrylamide gel electrophoresis followed by Coomassie blue staining.

## AlphaFold-multimer prediction methods

Predictions were generated with AlphaFold-Multimer[29,30] version 2.3.2 following a published approach[31] and using a computing cluster with these key settings:

    --db_preset=full_dbs  --max_template_date=2020-05-14  --models_to_relax=best  --model_preset=multimer  --num_multimer_predictions_per_model=5

The resulting predicted models were aligned in PyMOL v2.5.4 to assess prediction convergence, and this software was used to prepare all structural figures. For UNK, the IDR segment, the C-terminal region, or the full-length sequences were provided. For the CCR4-NOT subunits, sequences corresponding to experimentally determined structures were used (PDB accession codes 4crv and 4c0d)[51,58].

## Mass photometry

Mass photometry of UNK, UNK$_C$, UNK$_{E6}$, UNK$_{E8}$, and CNOT$_{MINI}$ was performed using the Refeyn TwoMP mass photometry instrument in buffer containing 50 mM HEPES, 200 mM NaCl, pH 7.5. Molecular weight calibrations were performed using two protein oligomer solutions, β-amylase (56, 112, and 224 kDa) and Thyroglobulin (670 kDa). The data acquisition was performed with AcquireMP (version 2023 R1.1) software and data analysis was performed with DiscoverMP (version 2023 R1.2) software.

## Individual-nucleotide resolution UV-crosslinking and immuno-precipitation (iCLIP)

All iCLIP experiments were performed in replicates following the iCLIP2 protocol[34]. Briefly, monoclonal HeLa cells inducibly expressing Flag-HA-tagged UNK$_{WT}$, UNK$_{dPAM2}$, or UNK$_{3M}$ were grown in 10 cm plates and harvested at 85% confluence. Prior to harvest, the cells were treated with Dox for 24 h or were left untreated. The cells were then washed with ice-cold PBS and irradiated with UV-light at 254 nm on ice. The irradiated cells were scraped, aliquoted into three 2-ml tubes, and

centrifuged at 5000 × g for 2 min at 4 °C. The supernatant was removed and the cell pellets were flash-frozen in liquid nitrogen and stored at −80 °C until use. Immunoprecipitation of the crosslinked UNK-RNA or PABPC1-RNA complexes was carried out using 2 μg of the anti-Flag antibody (Millipore Sigma, F1804, multiple lots) or anti-PABPC1 antibody (Abcam, ab21060, multiple lots). The complete iCLIP experiment, including deep sequencing of the prepared cDNA libraries, was repeated in four and two replicates for UNK and PABPC1 iCLIP libraries, respectively.

## RNA-seq library preparation

Total RNA from aliquots of samples used for ribosome profiling experiments (see Ribosome profiling) of each uninduced cells or cells expressing Flag-HA-tagged UNK$_{WT}$, UNK$_{dPAM2}$, or UNK$_{3M}$ for 24 h was extracted using Direct-zol RNA miniprep kit (Zymo Research, R2050) according to the manufacturer's instructions. Poly-A containing RNA was enriched from the total RNA using the NEBNext Poly(A) mRNA Magnetic Isolation Module (New England Biolabs, E7490S) and sequencing libraries were prepared using the NEBNext Ultra II Directional RNA Library Prep with Sample Purification Beads (NEB E7765S). RNA-seq libraries for each sample type were prepared, sequenced, and analyzed in triplicates.

## Ribosome profiling

Ribosome profiling experiments with uninduced cells or cells expressing Flag-HA-tagged UNK$_{WT}$, UNK$_{dPAM2}$, or UNK$_{3M}$ for 24 h were carried out in duplicates essentially as described[70] and by following the cDNA library-making protocol as for the iCLIP experiments[34]. Briefly, cells were grown in 15 cm dishes and harvested at 70% confluence. Prior to harvest, cells were treated or not with Dox for 24 h to induce the expression of the transgenes. Cells were then washed in ice-cold PBS, lysed for 10 min on ice in a lysis buffer, triturated by passing twice through a syringe fitted with a 26-gauge needle, and spun at 20,000 × g for 10 min at 4 °C. Cell lysates were digested with RNase I for 45 min at room temperature followed by the addition of SUPER-ase•In RNase Inhibitor (Thermo Fisher Scientific, AM2696). The lysates were underlaid with 1M sucrose and spun in a 50.4 Ti rotor at 311,643 × g for 2 h at 4 °C. Pellets were resuspended in TRIzol reagent (Life Technologies, 15596018) and RNA was extracted according to the manufacturer's instructions. The extracted RNA was size-selected by denaturing PAGE, retaining only fragments between 26 and 34 nts, and 3′ end-dephosphorylated with T4 PNK for 30 min at 37 °C followed by ligation to a pre-adenylated linker (L3-App) as described for the iCLIP procedure[34]. Unligated 3′ linker was removed by incubating the samples with the 5′-deadenylase KmHnt3 (see Recombinant protein expression and purification) and RecJ exonuclease (New England Biolabs, M0264S) for 45 min at 37 °C. The 3L-App ligated RNA was purified with Oligo Clean & Concentrator (Zymo Research, D4060), reverse transcribed by SuperScript IV Reverse Transcriptase (Thermo Fisher Scientific, 18090010), and converted to cDNA libraries for high-throughput sequencing, as described[34].

## Poly(A) tail length analysis

Total RNA was extracted from uninduced cells or cells expressing Flag-HA-tagged UNK$_{WT}$, UNK$_{dPAM2}$, or UNK$_{3M}$ for 24 h using Direct-zol RNA Miniprep (Zymo Research, R2050) according to the manufacturer's instructions. RNA quality was assessed using the 2100 Bioanalyzer (Agilent Technologies) with the RNA integrity number ranging from 8.4 to 10.

Libraries for direct RNA sequencing were prepared from mRNA in duplicates using Library Kit SQK-RNA002 (Oxford Nanopore Technologies) and sequenced on the GridION or PromethION 2 Solo device (Oxford Nanopore Technologies) using FLO-MIN106D or FLO-PRO002 flow-cells, respectively. One flow-cell was used for each sample.

## Quality control of cDNA libraries and High-throughput sequencing

iCLIP, RNA-seq, and ribosome profiling cDNA libraries were analyzed by non-denaturing PAGE and the 2100 Bioanalyzer (Agilent Technologies), quantified with the Qubit 2.0 Fluorometer (Life Technologies), pooled by library type, and sequenced using the HiSeq 2500, HiSeq 4000, or NovaSeq 6000 systems (all Illumina).

## Computational analyses

**Conservation and disorder score calculation.** Amino acid conservation of UNK was calculated using https://consurf.tau.ac.il, applying default settings. The disorder confidence score was calculated using the DISOPRED3 algorithm with default settings on the full-length mouse UNK amino acid sequence on the PSIPRED server (http://bioinf.cs.ucl.ac.uk/psipre).

**Analysis of iCLIP data.** The iCLIP data were processed essentially as described previously[71]. Briefly, data was assessed with FastQC (v0.11.9, https://www.bioinformatics.babraham.ac.uk/projects/fastqc/). High-quality data was chosen with fastq_quality_filter from FASTX Toolkit (v0.0.13, http://hannonlab.cshl.edu/fastx_toolkit/), with these parameters: -Q 33 -q 10 -p 100. The indexed sequencing reads were demultiplexed with flexbar (v3.5.0)[72,73] and then mapped to UCSC hg38 genome with STAR genome aligner(v2.7.3a)[74] using these parameters: --outFilterMismatchNoverReadLmax 0.04 --outFilterMismatchNmax 999 --outFilterMultimapNmax 1 --alignEndsType Extend5pOfRead1 --sjdbGTFfile gencode.v35.annotation.gtf --sjdbOverhang 75 --outReadsUnmapped Fastx --outSJfilterReads Unique --readFilesCommand zcat --outSAMtype BAM SortedByCoordinate --runThreadN 8. PCR duplicates were removed with umi_tools (v1.0.1)[75]. PureCLIP (v1.3.1, parameter: -ld -nt 8)[76] was utilized to identify individual crosslink events and for calling of peaks, i.e., binding sites. A minimum of 20 crosslink events were required for each peak. All peaks called by PureCLIP were expanded to a 9-nt region and assigned to Gencode (v35) comprehensive gene annotation[77]. For assessing the genomic distribution of iCLIP crosslink nucleotides, we used the following hierarchy: ncRNA > CDS > 3′UTR > 5′UTR > intron > other > intergenic (Supplementary Fig. 6E). Peaks mapping to different isoforms of a gene were assigned to the gene.

The classification of gene-binding strength (high, moderate, or low/no; Fig. 6, Supplementary Fig. 8) was based on the total tag number in all gene's peaks or on maximal tag number of individual, ~9-nt peaks. In particular, we classified as highly bound those genes that were either among the top 25% in their tag number within the called peaks or had a maximal peak height that ranked among the top 25% of all genes. We also classified as lowly or non-bound those genes whose both total tag number and maximal individual peak height ranked in the bottom 25% of genes in the respective categories. All other genes were classified as moderately bound.

The sequence composition at $UNK_{WT}$, $UNK_{dPAM2}$, or $UNK_{3M}$ binding sites was assessed as described previously (Fig. 5A, B)[17]. First, we identified the position of the maximum within each binding site (i.e., the nucleotide with the highest number of crosslink events; the first was taken in case of multiple nucleotides with equal counts) and extracted an extended window of 51 nts on either side. We counted the frequency of all 64 possible trinucleotides (triplets) at each position across all binding sites, counting each triplet on the first of three nucleotides. To correct for different background levels, we further normalized the frequency profile of each triplet to its median frequency across the complete 103-nt window, generating enrichment scores.

To compare the spatial arrangement of different triplets, we performed unsupervised hierarchical clustering of the normalized triplet profiles in a 31-nt window around the binding-site maxima of $UNK_{WT}$, $UNK_{dPAM2}$, or $UNK_{3M}$ (Fig. 5A; the heatmap profiles the enrichment scores). The resulting dendrogram was split into subtrees to obtain three sets of triplets with similar spatial distribution: (1) UAG, (2) U/A-rich triplets (UUU, AUU, UUA, CUU, UUC, UUG, AAU, AAA, GUU, UAA, UAU, and AUA), and (3) all remaining triplets. Triplet frequencies in each set were combined into a summarized profile (Fig. 5A, top).

To assess the prevalence of the UAG triplet in WT-specific and mutant-specific binding sites (Fig. 5E), the ratio of respective peaks with UAG within 15 nts upstream of the binding-site maxima versus all analyzed peaks was calculated. The occurrence of PABPC1 iCLIP peaks in the vicinity of $UNK_{WT}$, $UNK_{dPAM2}$, or $UNK_{3M}$ iCLIP peaks on mRNA (Supplementary Fig. 6K), the UNK-binding sites on mRNA were slopped with bedtools to upstream and downstream for 20 nt[78]. The overlapping PABPC1 peaks were counted with the bedmap[79]. An intersection of more than 1 nt was considered as overlapping.

**Ribosome profiling data analysis.** After quality control with FastQC (v0.11.9, https://www.bioinformatics.babraham.ac.uk/projects/fastqc/), sequencing reads were demultiplexed with flexbar (v3.5.0)[72,73] and mapped to the human rRNA with bowtie2 (v2.4.5)[80]. Hg38 rRNA sequences were retrieved from UCSC repeatmask database using table browser tool (https://genome.ucsc.edu/cgi-bin/hgTables). Reads not mapping to human rRNA were then mapped to hg38 lncRNA (Gencode v35; https://www.gencodegenes.org/human/) and the unmapped reads were aligned to hg38 protein-coding transcripts, keeping only uniquely mapping reads. Ribosome protected fragments (RPFs) in each gene (Gencode v35) were counted with samtools (v1.14)[81]. The differential ribosome occupancy was performed with DESeq2 package (v1.38.2)[82]. Because UNK targets the majority of all expressed transcripts, as indicated by the iCLIP analysis, instead of using DESeq2-inherent normalization, we normalized RPFs to 311 highly expressed genes (RPKM > 10) that had no $UNK_{WT}$, $UNK_{dPAM2}$, or $UNK_{3M}$ binding sites. We normalized RPF counts for individual genes using the formula 40,000*R/N, where 40,000 is an arbitrary number that is close to the average RPF count for the 311 genes across all conditions, N is the total RPF count for the 311 genes, and R is the RPF count for an individual gene. We considered as differentially translated genes with a $P$ value < 0.05 and fold change in ribosome occupancy > 2. The volcano plots in Fig. 7 and Supplementary Fig. 8 were drawn with ggplot2 package (ggplot2_3.4.0).

**RNA-seq data analysis.** The RNA-seq data were mapped to the human genome (hg38) using STAR (v2.7.3a). Read counts in each gene (Gencode v35) were calculated with featureCounts tools (v2.0.0) from the Rsubread package[83]. Differentially expressed genes (DEGs) were identified with DESeq2 (v1.38.2)[82]. Similar to the ribosome profiling data analysis, reads uniquely mapping to protein-coding genes were normalized to the 311 highly expressed, UNK-unbound genes. DEGs were chosen based on the adjusted P value threshold of 0.05 and fold change in expression > 2. RPKM values were calculated with cufflinks (v2.2.1)[84].

**Poly(A) tail-seq data analysis.** Raw reads from the poly(A) tail-seq libraries were base-called with guppy (v6.3.7, https://community.nanoporetech.com/downloads). Passing reads were mapped to Gencode v35 transcripts using minimap2[85,86]. Poly(A) tail length for each read was estimated using the Nanopolish (https://github.com/jts/nanopolish) and only length estimates with the QC tag reported as PASS were considered in subsequent analyses. Read numbers for genes with different transcript isoforms were combined. Reads per million (RPM) values were calculated and correlated with transcript abundance (RPKM; RNA-seq data; Supplementary Fig. 7A) and average tail length (Supplementary Fig. 7B). Tail length distributions were determined for different gene/transcript groups based on the numbers UNK-binding sites (BS): No BS, 1-2 BS, 3-10 BS, 11-30 BS, and 31 or more BS.

## Statistics and reproducibility

In Fig. 5B, the statistical significance for all pairwise comparisons for each set of trimers was determined using Wilcoxon signed rank exact test. The obtained $p$ values were as follows: p(UAG; WT-3M) = 3.05 × $10^{-5}$, $p$(UAG; WT-dPAM) = 3.05 × $10^{-5}$, $p$(UAG; 3M-dPAM) = 0.0003, $p$(U/A rich; WT-3M) = 1.19 × $10^{-5}$, $p$(U/A rich; WT-dPAM2) = 2.99 × $10^{-6}$, $p$(U/A rich; 3M-dPAM2) = 0.76, $p$(other; WT-3M) = 8.02 × $10^{-6}$, p(other; WT-dPAM2) = 5.61 × $10^{-5}$, p(other; 3M-dPAM2) = 0.42. In Fig. 5F, the statistical significance for all pairwise comparisons for each indicated segment of mRNA was determined using Wilcoxon rank sum test with continuity correction. The obtained $p$ values were as follows: $p$(5′UTR; WT-3M) = 0.33, $p$(5′UTR; WT-dPAM) = 0.0005, $p$(5′UTR; 3M-dPAM) = 0.0016, $p$(CDS; WT-3M) = 7.36 × $10^{-17}$, $p$(CDS; WT-dPAM2) = 4.61 × $10^{-19}$, $p$(CDS; 3M-dPAM2) = 1.93 × $10^{-5}$, $p$(3′UTR; WT-3M) = 3.84 × $10^{-7}$, $p$(3′UTR; WT-dPAM2) = 1.10 × $10^{-20}$, $p$(3′UTR; 3M-dPAM2) = 1.24 × $10^{-10}$.

In Fig. 6A–C, E–G, the statistical significance was determined using the Wald test with (Fig. 6A–C) or without (Fig. 6E–G) Benjamini–Hochberg adjustment for multiple comparisons.

In Fig. 6I, the box in each box plot represents the interquartile range (IQR) of the data, with the line in the middle of the box representing the median value (the 50th percentile), the lower part the first quartile (lower bound is the 25th percentile; Q1), the upper part the third quartile (upper bound is the 75th percentile; Q3). The upper or lower whiskers in each box plot extend from the upper bound toward the maximum (no further than Q3 + 1.5 × IQR) or from the lower bound toward the minimum value (no further than Q1 − 1.5 × IQR), respectively. Outliers are omitted.

## Reporting summary

Further information on research design is available in the Nature Portfolio Reporting Summary linked to this article.

# Data availability

The data supporting the findings of this study are available from the corresponding authors upon request. The high-throughput sequencing data generated in this study have been deposited in the Gene Expression Omnibus (GEO) database under accession code GSE240571. The mass spectrometry proteomics data have been deposited in the ProteomeXchange Consortium via the PRIDE[87] partner repository with the dataset identifier PXD050601. Source data for the figures and Supplementary Figs. are provided as a Source Data file. Source data are provided with this paper.

# Code availability

The original code used to analyze the data and generate figures is available at https://github.com/Shiyang-He/Unkempt-Project-data-analysis/ or at Zenodo: https://doi.org/10.5281/zenodo.10783732[88].

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

## Acknowledgements

The authors thank M. Oakes, K. Jepsen, S. E. O'Leary, Y. Kanomata, S.-A. Chung, R. Tomaino, Y. Wu, and C. Cai for assistance and helpful discussions; F. Lan for the pDEST-pcDNA3-Gal4 plasmid; N. Ingolia for the pNTK576-pET28a-His6x-KmHnt3 plasmid; A.B. Shyu for the anti-CNOT7 antibody; the UCI Genomics Research & Technology Hub, the UCSD IGM Genomics Center, the Taplin Mass Spectrometry Facility at Harvard Medical School, and the Biophysics Resource in the Center for Structural Biology, Center for Cancer Research, NCI. This work was supported in part by the U.S. National Institutes of Health grant R01 GM144693 (to J.M.). D.D. was supported by the U.S. National Institutes of Health grant R35 GM142864. J.B. was supported a grant from the USA Department of Defense (W81XWH-17-1-0082). S.C. was supported by the U.S. National Institutes of Health grant R35 GM151004. D.J.T., J.C., K.R., and E.V. were supported by the Intramural Research Program, Center for Cancer Research, National Cancer Institute, National Institutes of Health.

## Author contributions

K.S., S.H., D.J.T., E.V., and J.M. conceptualized the study and designed experiments and data analyses. K.S., D.J.T., K.R., J.C., and J.M. performed all experiments. S.H. and E.V. performed bioinformatic analyses. D.D., J.M.B., S.C., and C.I. contributed key reagents and provided guidance. E.V. and J.M. acquired funding and supervised this work. J.M. wrote the manuscript with assistance from D.J.T. and E.V. and input from all other authors.

## Competing interests

The authors declare no competing interests.
