## [Peer Review File · Nature Communications]

Regulation by the RNA-binding protein Unkempt at its effector interfaceREVIEWER COMMENTS

Reviewer #1 (Remarks to the Author):

Summary

In this manuscript, the authors boldly propose “a paradigm” for the regulations and functions of RBPs at the effector interface. They took the developmental gene *Unkempt* (UNK) as a model RBP and identified the CCR4-NOT complex and PABPC as crucial effectors for it. They discovered that the IDR region of UNK has regulatory functions by providing multivalent interaction interfaces for different effectors such as the CCR4-NOT complex and PABPC. Moreover, the authors precisely mapped the interfaces at the single amino acid level using biochemical and structural modeling, and further addressed that the interaction with the CCR4-NOT complex is critical for the function of UNK in translational suppression of its target genes and cell polarization.

While the biochemical data is very convincing and provides clear molecular details of the protein-protein interaction between UNK and its effectors such as CCR4-NOT complex and PABPC, the conclusions and interpretations are often overemphasized from weak experimental evidence, especially throughout Figure 6-7. For example, the enrichment of different motifs in iCLIP-seq peaks between WT and mutants seems only slightly different, questioning whether there's any statistical significance. Therefore, I would suggest the authors to tone-down the conclusions and emphasize only the most compelling data in the manuscript, which I believe will be still interesting enough for the broad readership of Nature Communications. Below are my suggestions for the improvements and clarification of the manuscript.

1. The use of the word ‘paradigm’ in the title and the manuscript text can be toned down without compromising the integrities of the conclusions. For example, “A model for regulation at the effector interface of a RNA-binding protein *Unkempt*” could be more specific and may well convey the significance of the study as well.
2. Although UNK is exclusively cytoplasmic (Fig S1B-C), the authors used GAL4-UAS transcriptional reporter as their main readout in Fig 1-2, which is active in the nucleus. How can the reporter be regulated by the cytosolic factors? Related to this, even though the authors showed that UNK mutants incapable of CCR4-NOT1 complex binding could not activate the reporter, this does not appear to be sufficient to prove the involvement of the CCR4-NOT1 complex. Can the authors provide direct experimental evidence that the CCR4-NOT1 complex or CNOT9 is involved in the regulation of the transcriptional reporter by using siRNAs or CNOT9 knockout cells as in Fig 3?
3. Alternative to point #2, since the main conclusion of the study is the translational regulation of UNK target genes by CCR4-NOT1 interaction, it would be nicer for the authors to utilize a translational reporter system (such as Lambda N or MS2 tethering) for the functional demonstration of effector interaction interfaces of IDR.
4. The interpretation of Fig 6A-B is not convincing. There are only very minor changes between WT and mutants. Are there any statistical significances? Similarly, for Fig 6E and 6F, are these results statistically significant? Even if the interpretations are correct, what would be the meaning of the results? What would be the functional outcomes of UNK re-localization to the coding region in the absence of PABPC binding in Fig 6F (by dPAM2 mutant)?
5. Can the downregulated genes upon UNK WT or mutant expression in Fig 7A-D be transcriptional or post-transcriptional? Or can these be secondary transcriptional targets of the UNK translational target genes shown in Fig 7E-G?

6. In lines 498-499, the authors claim that “the UNK-CCR4-NOT nexus is a critical conduit of translational repression for a large fraction of the cellular mRNA pool”. Could this be confirmed by genetic rescue experiments using CNOT9 siRNA or knockout cells? For example, the authors can perform ribosome profiling in CNOT9 knockout cells with UNK wt overexpression to see the rescue of translational suppression of UNK target genes in CNOT9 KO cells. Alternatively, using naturally UNK-expressing cells (embryonic or neuronal cells), the authors can perform ribosome profiling after CNOT9 knockdown to see that endogenous UNK target genes are translationally suppressed in a CNOT9-dependent manner.

7. For the clarification of the model: It is not yet experimentally demonstrated whether the CCR4-NOT1 complex and PABPC interact with UNK simultaneously to form a single large multi-complex protein as depicted in Fig 7J or Fig S7. Can the authors at least perform mass photometry analysis (as in Fig 5) to demonstrate this?

8. The statistical significances should be compared between si-Control to other siRNAs in Fig 3E. Likewise, the significance mark in Fig 3G should also be compared between WT (+/+) and the others. In Fig 3E and 3G, UNK WT expression should be indicated.

9. It was previously shown that the CCR4-NOT1 complex cooperates with PABP to promote mRNA deadenylation (Yi et al., 2018, doi: 10.1016/j.molcel.2018.05.009). Considering the current manuscript, UNK may compete with the deadenylation machinery composed of CCR4-NOT1-PABP, possibly in the embryonic context. Please comment on this possibility in the discussion (lines 586-592).

Reviewer #2 (Remarks to the Author):

Shah et al. in the work entitled ‘A paradigm for regulation at the effector interface with RNA-binding proteins’ characterize the UNK protein Intrinsically Disordered Region (IDR) and show that it is divided into three functional regions: PAM-2 that binds to polyA-binding protein PABPC and two separate regions that collectively bind to NOT-module and CNOT9 subunits of the CCR4-NOT complex. Biochemical data was evidenced by mutational analysis of the UNK protein in eukaryotic cell lines, using recombinant proteins along with protein-protein interaction modeling. The simultaneous UNK binding to the large CCR4-NOT and PABPC protein is seemingly facilitated by the dimerization capacity of UNK, also extensively characterized. Authors further describe the importance of the UNK-CCR4-NOT-PABPC interaction. First, they show that CCR4-NOT subunits, other than the catalytic ones are required for establishment of the cellular axial ratio; a phenotype characteristic to UNK expression, and that loss of UNK IDR elements that recruit CCR4-NOT and PABPC has a similar effect. The key element of the publication, that inspired its title, is that binding of UNK to CCR4-NOT complex and PABPC can influence the selection of UNK protein binding sites on diverse mRNAs, which was done by iCLIP. Though these changes on a motif level are actually subtle, it is an important concept that was somehow present in the community as theoretical consideration, but never really proven. In this sense the paper is quite important. In the subsequent sections the authors show that UNK does not influence the deadenylase activity of CCR4-NOT, but instead heavily influences ribosome mRNA occupancy, thus regulating gene expression.

Publication by Shah et al. is an impressive amount of work presenting meticulous investigation of a molecular protein-protein interaction interface with its putative functional significance. While the protein-protein interaction is exhaustingly characterized, the functional significance of this interaction; in particular for the control of ribosome occupancy; opens more questions than it answers. However, given the impressive amount of biochemical data presented and the fact that it also brings resource data that will be crucial for further functional analyses, effectively paving the way for a potentially great follow-up publication, I think that the publication should be accepted with minor revisions. I

congratulate the authors for this work.

Major comments:

Given the unaltered interaction of UNK with PABPC in condition of mutation of the CCR4-NOT-binding module and the postulated influence of PABPC on UNK distribution, what is the effect of PABPC Kd on axial ratio and UNK transcriptional activity? The UNK-PAMdelta was assessed but why wasn't PABPC KD investigated like CNOT9?

Question related to 1G -> what is the morphology of cells expressing the IDR fragments alone? Is the IDR sufficient to induce cell morphology switch?

Figure 3A,B and H -> those are results of a BioID experiments. The authors zoom into the fraction of proteins enriched in the UNK WT or mutant IPs. The entire plot looks like there were virtually no proteins detected in the background purification. While I understand that this was done for the sake of the clarity of the Figure, the end result is that the raw data is hidden. Likewise, one cannot appreciate if Ccr4-NOT and PABPC were the most enriched proteins in the BioID or not. If the authors should like to maintain this display I think that they should also show the raw Bio-ID volcano plot in the supplemental information.

After snailing through the biochemical part of the publication, the general feeling is that the functional part is designed to be a quick shuttle to the discussion. A few more words and single gene examples would be much helpful. Figure 7 is composed almost uniquely of volcano plots of RNA-seq and ribosome profiling data, which are a great way to give a first impression of the raw data and in most cases are put in the supplement. The comparisons of those data are only given in the main text. I would ask the authors to think about a better representation of the raw data. For instance a scatterplot (or density plot ?) that would compare the fold-change of mRNA level on x-axis and the fold-change in ribosome occupancy on y-axis followed by a Venn digram would be much more informative. Such representation could include all mRNAs as a density plot and those that are significantly changed as dots. That done for each mutant. Also for this section it would be nice to show single gene examples, which were evidenced in Fig. 6 (S100A11, RPL30 and HINT1) to show ribosome occupancy change?

The biggest question that arises from the results is the mechanism by which UNK and its interaction with CCR4-NOT and PABPC regulates ribosome occupancy. It is striking that the 3M mutant actually displays less phenotype regarding ribosome occupancy than the PAM-2delta. This could suggest that the main UNK effector in this circuit is actually PABPC. Some suggestions exist that in various organisms PABPC interaction with the cap binding complex is relevant to translation initiation (doi: 10.4161/2169074X.2014.959378; <https://doi.org/10.1074/jbc.M602780200>). Did the authors see any UNK interaction with cap binding complex? Could they perhaps better indicate whether simultaneous cap-binding complex-UNK-PABPC interaction is at all possible? If that is actually the main subject of the follow-up publication I would be satisfied with a more general indication.

Minor points:

Publication title -> might be better to actually include UNK in the title. I understand that the general concept needs to be promoted but the specific case is also important.

Fig. 1F -> I find the y-axis somewhat unclear. And would therefore ask for clarification in the caption or on the main figure: What is the exact difference between samples marked 1, 2, 3... 11, 12, 13, 14, 15 etc. and UNK(d11), UNK(d14)? I imagine it is (1.) the fragment alone or (2.) the entire protein with the indicated fragment deleted? If so the greek symbol would be best.

Fig. 1F and 1G -> match the order of samples for better clarity.

Section title ,Unkempt interacts directly with multiple CCR4-NOT complex subunits' - is somehow confusing. Maybe better to make it more precise? - ,Unkempt interacts directly with CCR4-NOT complex core NOT module and CNOT9, but not with the catalytic module?'

THERE IS NO MESSAGE TO THE EDITOR THAT IS HIDDEN FROM THE AUTHORS.

Reviewer #3 (Remarks to the Author):

How RBPs select and regulate target mRNAs is a difficult problem due to many reasons including transient binding, involvement of IDRs, and ambiguous binding specificity. The authors chose an RBP (Unkempt) that regulates target mRNA by translational inhibition and is crucial for cellular polarization and performed extensive biochemical study, structure prediction, and systems analyses to understand its molecular mechanism of target selection. The biochemical study using various constructs and mutants of Unkempt and effector proteins and AlphaFold-predicted structural models is impressive. The proposed binding model of Unkempt and effectors makes sense. Although the issue of the transcriptional activity was not clarified and the target selection mechanism was not completely explained by the current binding model, the study substantially improved our understanding of how RBPs could work on target RNA. I have only a few minor comments, but they need to be addressed before publication.

1. This study focuses on a single RBP. No evidence is provided to claim a general mechanism for RBPs. This study could be a framework for the future studies on the functional mechanisms of other RBPs but not for understanding them. The authors need to avoid generalization through Abstract and Main text.
2. The authors need to consider moving the study on transcriptional activity to the back and adding discussion why transcriptional and translational activities are difficult to dissect and how they would be coupled. Would it be possible that Unkempt-bound mRNA is transported back into the nuclear?
3. The authors need to describe what d14, d11, d15, and d17 indicates in the main text and figure legend 1. I assumed that they indicate the deletion of constructs 14, 11, 15, and 17.
4. Why negatively charged residues were chosen for the mutations V511E, I515E and L522E? These changes may have an effect on RNA-binding through the interaction with neighboring positively charged residues.
5. line 220, this sentence is unclear.
6. line 302, What is "d positions?"

REVIEWER COMMENTS

Reviewer #1 (Remarks to the Author):

Summary

In this manuscript, the authors boldly propose “a paradigm” for the regulations and functions of RBPs at the effector interface. They took the developmental gene Unkempt (UNK) as a model RBP and identified the CCR4-NOT complex and PABPC as crucial effectors for it. They discovered that the IDR region of UNK has regulatory functions by providing multivalent interaction interfaces for different effectors such as the CCR4-NOT complex and PABPC. Moreover, the authors precisely mapped the interfaces at the single amino acid level using biochemical and structural modeling, and further addressed that the interaction with the CCR4-NOT complex is critical for the function of UNK in translational suppression of its target genes and cell polarization.

While the biochemical data is very convincing and provides clear molecular details of the protein-protein interaction between UNK and its effectors such as CCR4-NOT complex and PABPC, the conclusions and interpretations are often overemphasized from weak experimental evidence, especially throughout Figure 6-7. For example, the enrichment of different motifs in iCLIP-seq peaks between WT and mutants seems only slightly different, questioning whether there’s any statistical significance. Therefore, I would suggest the authors to tone-down the conclusions and emphasize only the most compelling data in the manuscript, which I believe will be still interesting enough for the broad readership of Nature Communications. Below are my suggestions for the improvements and clarification of the manuscript.

We thank the Reviewer for the favorable assessment and constructive criticism of our study. Below, we address the Reviewer’s comments point-by-point.

1. The use of the word ‘paradigm’ in the title and the manuscript text can be toned down without compromising the integrities of the conclusions. For example, “A model for regulation at the effector interface of a RNA-binding protein Unkempt” could be more specific and may well convey the significance of the study as well.

We agree and have modified the title of the manuscript to "Regulation by the RNA-binding protein Unkempt at its effector interface". We also replaced the word 'paradigm' with 'model' where used in the manuscript (*lines 82 and 350*). In addition, we toned down several of our interpretations and removed generalizations of our findings, as recommended. We trust that the revised manuscript appropriately discusses that significance of our findings for regulation by Unkempt while avoiding generalizations.

2. Although UNK is exclusively cytoplasmic (Fig S1B-C), the authors used GAL4-UAS transcriptional reporter as their main readout in Fig 1-2, which is active in the nucleus. How can the reporter be regulated by the cytosolic factors? Related to this, even though the authors showed that UNK mutants incapable of CCR4-NOT1 complex binding could not activate the reporter, this does not appear to be sufficient to prove the involvement of the CCR4-NOT1 complex. Can the authors provide direct experimental evidence that the CCR4-NOT1 complex or CNOT9 is involved in the regulation of the transcriptional reporter by using siRNAs or CNOT9 knockout cells as in Fig 3?

We note that while the bulk of the UNK protein is indeed cytoplasmic, a small fraction of overexpressed or endogenous UNK is consistently nuclear (Supplementary Fig. 1; see also Murn et al, 2015, PMID: 25737280). While this may suggest a low level of nuclear import and a potential transcriptional role of UNK in cells, we carried out several dedicated experiments but failed to find any evidence for such activity, as we describe in the manuscript (*lines 116-122*). Nevertheless, the small amount of nuclear UNK and forced recruitment of UNK to the promoter region of the luciferase reporter plasmid, via fusion with the Gal4 domain, may explain the observed reporter-linked transcriptional activity.

Our mutagenetic analysis led us to identify some of the critical residues in UNK_{IDR} that are essential for UNK’s transcriptional activity and, as we demonstrate in subsequent experiments, also mediate interactions with the CCR4-NOT complex. Although this opens up a possibility that the CCR4-NOT complex may be required for UNK’s transcriptional activity, this is not necessarily so. To probe this possibility, we carried out the transcriptional reporter assay in CNOT9 knockout cells, as suggested by the Reviewer. We found that the Gal4-UNK fusion was comparably transcriptionally active in the CNOT9 knockout as it was in wildtype cells (**Fig. R1**). This result indicates that, unlike UNK’s translational control, the reporter-linked transcriptional activity of UNK does not rely on the CNOT9 subunit. It

is conceivable that, in the absence of CNOT9, the CCR4-NOT complex is efficiently recruited to the promoter-tethered UNK via its NOT module to drive transcription of the reporter gene. Alternatively, transcriptional effectors other than CCR4-NOT might interact with the critical residues of UNK in the chromatin environment. We have included these comments in the discussion of the revised manuscript (*lines 439-445*).

Fig. R1. Transcriptional activity of UNK measured by a dual luciferase reporter assay at 24 h after transfection of HeLa cells with constructs for expression of the indicated, Gal4 DNA-binding domain (Gal4)-tagged proteins. -, Gal4 alone; RLU, relative luminescence units; WT, wildtype cells; CNOT9^{KO}: CNOT9 knockout cells. Data are presented as mean \pm SD. Statistical significance was determined using Student's t-test (n = 6).

3. Alternative to point #2, since the main conclusion of the study is the translational regulation of UNK target genes by CCR4-NOT1 interaction, it would be nicer for the authors to utilize a translational reporter system (such as Lambda N or MS2 tethering) for the functional demonstration of effector interaction interfaces of IDR.

Our goal in this study was to investigate RNA regulation at the effector interface of UNK in its natural form, bound to its targets at the endogenous mRNA binding sites in cells. We employed approaches, including ribosome profiling, iCLIP, RNA-seq, and poly(A)tail-seq, that allowed us to do so in an unperturbed setting transcriptome-wide. We demonstrated the functional involvement of the UNK-effector interface in translational regulation in a quantitative manner by employing these approaches in conjunction with targeted mutagenesis of UNK and, independently, genetic perturbation of the key effector subunit, CNOT 9. While translational reporter assays are convenient, we avoided their use given that these assays require interference with the native RBP-RNA binding, altering the native binding affinity and position of RNA binding. In reporter assays, an RBP is typically tethered to 3' UTR of the reporter mRNA, whereas UNK exhibits a composite binding pattern, with much of it bound to CDS. This observation and our finding that UNK mutants show distinct binding patterns suggest that the position of RNA binding might also regulate the function of UNK. It is plausible that this effect would be lost by tethering UNK to RNA. We also reasoned that reporter systems may inadvertently perturb RBP-effector interactions, which, in the case of the UNK-CCR4-NOT interface, are highly complex and thus likely sensitive to manipulation. These considerations prompted us to study the effector interface of UNK in its native context in cells.

4. The interpretation of Fig 6A-B is not convincing. There are only very minor changes between WT and mutants. Are there any statistical significances? Similarly, for Fig 6E and 6F, are these results statistically significant? Even if the interpretations are correct, what would be the meaning of the results? What would be the functional outcomes of UNK re-localization to the coding region in the absence of PAPBC binding in Fig 6F (by dPAM2 mutant)?

We analyzed the statistical significance of the results showing the impact of effector interactions on RNA binding by UNK, as suggested (original Fig. 6, new Fig. 5). We found that both UNK recognition motifs, UAG and the U/A-rich motif, were significantly enriched within binding sites of UNK_{WT} compared to its mutants, UNK_{dPAM2} or UNK_{3M}, whereas all other motifs were significantly depleted ($p < 6 \times 10^{-5}$; original Fig. 6A, B, new Fig. 5A, B). Analysis of the data in the original Fig. 6E (new Fig. 5E) revealed a small but significant difference in the occurrence of the UAG trimer within 15 nts upstream of the WT-specific or mutant-specific peaks ($p < 0.05$). The differences in distribution of UNK on mRNA were highly significant for any pairwise comparison between UNK_{WT}, UNK_{dPAM2}, and UNK_{3M} within coding sequences or 3' UTRs ($p < 2 \times 10^{-5}$; original Fig. 6F, new Fig. 5F). We have included these results in the legend to Fig. 5 in the revised manuscript.

Our documented impact of effector interactions on RNA binding by UNK provide the first experimental support for an important concept that, as Reviewer #2 writes, “was somehow present in the community as theoretical consideration, but never really proven.” The identification of CCR4-NOT and PABPC (effectors) as regulators of RNA binding by UNK may have important regulatory implications. For instance, loss of UNK–effector interactions could contribute to the observed targeting and potential regulation of several hundred mRNAs by UNK_{dPAM2} and/or UNK_{3M} that are [not bound by UNK_{WT} and, vice-versa, result in loss of binding and regulation of numerous mRNAs that are typically bound by UNK_{WT} (Supplementary Fig. 6G). Likewise, securing UNK to 3'UTR and limiting its localization to the coding region, especially by PABPC, may contribute to the translational repression by UNK, although the mechanism of such action remains unclear (Supplementary Fig. 5F, 6E, 6F). We envision that the effector-controlled distribution of UNK on mRNAs might also play an indirect role, for instance, by regulating access of other RBPs to the targeted mRNAs.

5. Can the downregulated genes upon UNK WT or mutant expression in Fig 7A-D be transcriptional or post-transcriptional? Or can these be secondary transcriptional targets of the UNK translational target genes shown in Fig 7E-G?

We believe that the bulk of the changes in gene expression detected by RNA-seq (original Fig. 7A-D, new Fig. 6A-D) are due to indirect effects of the translational repression by UNK. A dominant direct effect of UNK-regulated transcription is unlikely because of 1) the largely cytoplasmic localization of UNK and our inability to detect endogenous transcriptional role for UNK in cells, as discussed in point #2, and 2) the stimulating effect of UNK on transcription detected by our luciferase reporter assay, which jars with the bias toward downregulation seen by RNA-seq. We expect that most of the observed changes in mRNA levels are caused indirectly because of the relatively weak binding of UNK to the highly regulated mRNAs and, vice versa, stronger binding to the weakly regulated mRNAs (iCLIP data in Fig. 6A-C). In addition, we suggest that the perturbation of the transcriptome is largely an indirect consequence of UNK's translational activity, given that UNK_{3M}, which is translationally essentially inactive, causes only minimal changes in mRNA levels (Fig. 6C, D, G, H), whereas UNK_{dPAM2} that exhibits intermediate translational repression elicits intermediate level of transcriptome perturbation (Fig. 6B, D, F, H). We leave open the possibility that the moderate bias towards downregulated mRNA levels observed with UNK_{WT} (Fig. 6A, D) is caused by a potential direct but mild mRNA-destabilizing effect of UNK. We agree with the Reviewer that the bias toward downregulated mRNAs could also be due to the secondary transcriptional targeting by the translational targets of UNK_{WT}.

6. In lines 498-499, the authors claim that “the UNK-CCR4-NOT nexus is a critical conduit of translational repression for a large fraction of the cellular mRNA pool”. Could this be confirmed by genetic rescue experiments using CNOT9 siRNA or knockout cells? For example, the authors can perform ribosome profiling in CNOT9 knockout cells with UNK wt overexpression to see the rescue of translational suppression of UNK target genes in CNOT9 KO cells. Alternatively, using naturally UNK-expressing cells (embryonic or neuronal cells), the authors can perform ribosome profiling after CNOT9 knockdown to see that endogenous UNK target genes are translationally suppressed in a CNOT9-dependent manner.

The suggested genetic rescue experiments using CNOT9 knockout cells were included in the original manuscript as a supplement and may have been missed by the Reviewer (original Fig. S6B-G, new Supplementary Fig. 8H-M). These experiments genetically validate the central role of the UNK-CCR4-NOT interface by showing that deletion of the CNOT9 subunit, much like mutations of UNK that disrupt the interaction with CCR4-NOT (Fig. 6G), essentially eliminate the UNK-mediated translational repression.

7. For the clarification of the model: It is not yet experimentally demonstrated whether the CCR4-NOT1 complex and PABPC interact with UNK simultaneously to form a single large multi-complex protein as depicted in Fig 7J or Fig S7. Can the authors at least perform mass photometry analysis (as in Fig 5) to demonstrate this?

We failed to detect a stable ternary complex of UNK, PABPC1, and CCR4-NOT by mass photometry. This technique relies on measurements in a very dilute concentration range (low nanomolar), and this concentration regime may lead to the dissociation of low-affinity interactions during the course of the measurement. We do, however, provide evidence in Supplementary Fig. 5D that stoichiometric amounts of PABPC1 and CNOT1/2/3/9 bound to UNK in isolation or in combination, suggesting that both PABPC1 and CNOT1/2/3/9 could bind UNK simultaneously.

8. The statistical significances should be compared between si-Control to other siRNAs in Fig 3E. Likewise, the significance mark in Fig 3G should also be compared between WT (+/+) and the others. In Fig 3E and 3G, UNK WT expression should be indicated.

We determined the statistical significance for the requested comparisons in the original Fig. 3E and 3G (new Fig. 2E and 2G). We now also indicate which samples in Fig. 2E and 2G express UNK_{WT} or its mutants.

9. It was previously shown that the CCR4-NOT1 complex cooperates with PABP to promote mRNA deadenylation (Yi et al., 2018, doi: 10.1016/j.molcel.2018.05.009). Considering the current manuscript, UNK may compete with the deadenylation machinery composed of CCR4-NOT1-PABP, possibly in the embryonic context. Please comment on this possibility in the discussion (lines 586-592).

Thank you for the suggestion. We have included this comment and the suggested citation in the discussion section (lines 433-435).

Reviewer #2 (Remarks to the Author):

Shah et al. in the work entitled 'A paradigm for regulation at the effector interface with RNA-binding proteins' characterize the UNK protein Intrinsically Disordered Region (IDR) and show that it is divided into three functional regions: PAM-2 that binds to polyA-binding protein PABPC and two separate regions that collectively bind to NOT-module and CNOT9 subunits of the CCR4-NOT complex. Biochemical data was evidenced by mutational analysis of the UNK protein in eukaryotic cell lines, using recombinant proteins along with protein-protein interaction modeling. The simultaneous UNK binding to the large Ccr4-NOT and PABPC protein is seemingly facilitated by the dimerization capacity of UNK, also extensively characterized. Authors further describe the importance of the UNK-CCR4-NOT-PABPC interaction. First, they show that CCR4-NOT subunits, other than the catalytic ones are required for establishment of the cellular axial ratio; a phenotype characteristic to UNK expression, and that loss of UNK IDR elements that recruit CCR4-NOT and PABPC has a similar effect.

The key element of the publication, that inspired its title, is that binding of UNK to CCR4-NOT complex and PABPC can influence the selection of UNK protein binding sites on diverse mRNAs, which was done by iCLIP. Though these changes on a motif level are actually subtle, it is an important concept that was somehow present in the community as theoretical consideration, but never really proven. In this sense the paper is quite important. In the subsequent sections the authors show that UNK does not influence the deaenylase activity of CCR4-NOT, but instead heavily influences ribosome mRNA occupancy, thus regulating gene expression.

Publication by Shah et al. is an impressive amount of work presenting meticulous investigation of a molecular protein-protein interaction interface with its putative functional significance. While the protein-protein interaction is exhaustingly characterized, the functional significance of this interaction; in particular for the control of ribosome occupancy; opens more questions than it answers. However, given the impressive amount of biochemical data presented and the fact that it also brings resource data that will be crucial for further functional analyses, effectively paving the way for a potentially great follow-up publication, I think that the publication should be accepted with minor revisions. I congratulate the authors for this work.

We thank the Reviewer for the positive evaluation and the compliments. We address the comments in the paragraphs below.

Major comments:

Given the unaltered interaction of UNK with PABPC in condition of mutation of the CCR4-NOT-binding module and the postulated influence of PABPC on UNK distribution, what is the effect of PABPC Kd on axial ratio and UNK transcriptional activity? The UNK-PAMdelta was assessed but why wasn't PABPC KD investigated like CNOT9?

We considered but chose to omit the proposed PABPC depletion experiment due to the following consideration. A number of human cell lines, including HeLa that we used in our study, express high levels of the PABPC1 (about 4 μ M in HeLa cells) and PABPC4 (about 0.9 μ M in HeLa cells) that contribute to the collective PABPC protein pool (Gorlach et al, 1994, PMID: 7908267; Hein et al, 2015, PMID: 26496610; Kajjo et al, 2022, PMID: 35156721). PABPC4 is about 80% identical to PABPC1 in amino acid sequence and the two proteins cooperate to support protein synthesis and cell viability. The groups of Nahum Sonnenberg and Marc Fabian demonstrated that depletion of either PABPC1 or PABPC4 alone had little effect on translation and cell viability, whereas co-depletion of PABPC1 and PABPC4 resulted in rapid cell death (Yoshida et al, 2006, PMID: 16601676; Kajjo et al, 2022, PMID: 35156721). Our own and published mass spec analyses point to a strong, PAM2-dependent interaction of UNK with PABPC1 or PABPC4 in cells (Fig. 2A, B, H, Supplementary Fig. 4A; Youn et al, 2018, PMID: 29395067), suggesting that the two PABPC proteins may substitute one another in their interaction with UNK. We reasoned that depletion of either of the PABPC proteins alone may exhibit little if any effect, whereas co-depletion of PABPC1 and PABPC4 would compromise cell viability and confound the measurement of UNK's transcriptional activity and cellular axial ratios,

rendering the results difficult to interpret. We thus considered the UNK_{dPAM2} mutant as the most reliable condition to prevent the interactions with either PABPC1 or PABPC4 and to allow for assessing the significance of these interactions in cells.

Question related to 1G -> what is the morphology of cells expressing the IDR fragments alone? Is the IDR sufficient to induce cell morphology switch?

We performed these experiments and found no changes in cell morphology upon expression of either the full-length IDR or its fragments. This suggests a mandatory participation of other segments of the UNK protein in its mediated cell morphogenesis.

Figure 3A,B and H -> those are results of a BioID experiments. The authors zoom into the fraction of proteins enriched in the UNK WT or mutant IPs. The entire plot looks like there were virtually no proteins detected in the background purification. While I understand that this was done for the sake of the clarity of the Figure, the end result is that the raw data is hidden. Likewise, one cannot appreciate if Ccr4-NOT and PABPC were the most enriched proteins in the BioID or not. If the authors should like to maintain this display I think that they should also show the raw Bio-ID volcano plot in the supplemental information.

The result shown in the original Fig. 3A (new Fig. 2A) is a reanalysis of BioID data reported by the Gingras lab (Youn et al, 2018, PMID: 29395067). We initially only plotted the 104 interacting proteins that passed the default threshold set by the authors for inclusion into their dotplots. We removed this score filtering setting and now show all 1483 interacting proteins identified by the N- and/or C-terminal fusions of BirA with UNK (BirA_UNK and UNK_BirA, respectively) as well as the complete set of raw data (Fig. 2A and Supplementary Table 2). The original Fig. 3B and H (new Fig. 2B and H) show our mass spectrometry analyses of tandem affinity purified protein complexes formed by UNK_{WT}, UNK_{3M}, and UNK_{dPAM2} (note that these were not BioID experiments). Here, we initially only showed proteins that were identified in the UNK complexes but not in the mock complex. We now show all proteins that were detected in the UNK_{WT}, UNK_{3M}, and UNK_{dPAM2} complexes, regardless of whether they were also found in the mock complex (Fig. 2B and H). We also include a listing of all proteins identified in all complexes, along with the numbers of detected unique and total peptides (Supplementary Table 2). Together, these results demonstrate that CCR4-NOT and PABPC1/4 interact strongly with UNK in cells.

After snailing through the biochemical part of the publication, the general feeling is that the functional part is designed to be a quick shuttle to the discussion. A few more words and single gene examples would be much helpful. Figure 7 is composed almost uniquely of volcano plots of RNA-seq and ribosome profiling data, which are a great way to give a first impression of the raw data and in most cases are put in the supplement. The comparisons of those data are only given in the main text. I would ask the authors to think about a better representation of the raw data. For instance a scatterplot (or density plot ?) that would compare the fold-change of mRNA level on x-axis and the fold-change in ribosome occupancy on y-axis followed by a Venn diagram would be much more informative. Such representation could include all mRNAs as a density plot and those that are significantly changed as dots. That done for each mutant. Also for this section it would be nice to show single gene examples, which were evidenced in Fig. 6 (*S100A11*, *RPL30* and *HINT1*) to show ribosome occupancy change?

This is a great suggestion. We chose volcano plots to present RNA-seq and ribosome profiling data primarily because this allowed us to easily correlate each dataset with the iCLIP results, thus facilitating visual distinction between direct and indirect effects of UNK (original Fig. 7A-G, new Fig. 6A-G). We now also include scatterplots that directly compare RNA-seq and ribosome profiling data for UNK_{WT}, UNK_{3M}, or UNK_{dPAM2} versus uninduced control, as well as the corresponding Venn diagrams to indicate the actual transcript numbers in each category (Supplementary Fig. 8B-G). The three transcripts shown in the original Fig. 6D (new Fig. 5D), *S100A11*, *RPL30*, and *HINT1*, were selected based on their strong but few UNK binding sites and the correspondingly low number of total UAG trimers per cDNA, to simplify visualization of the distinctive RNA binding pattern of UNK mutants. We note that these transcripts do not show significant differences in their ribosome occupancy in UNK-expressing cells and we therefore did not highlight them in the scatterplots. However, we provide tables with the complete set of analyzed data that can be browsed to find transcripts with defined changes in ribosome occupancy, levels of expression, and/or strength of binding by UNK (Supplementary Tables 3, 5, 7).

The biggest question that arises from the results is the mechanism by which UNK and its interaction with CCR4-NOT and PABPC regulates ribosome occupancy. It is striking that the 3M mutant actually displays less phenotype regarding ribosome occupancy than the PAM-2delta. This could suggest that the main UNK effector in this circuit is actually PABPC. Some suggestions exist that in various organisms PABPC interaction with the cap binding complex

is relevant to translation initiation (doi: 10.4161/2169074X.2014.959378; <https://doi.org/10.1074/jbc.M602780200>). Did the authors see any UNK interaction with cap binding complex? Could they perhaps better indicate whether simultaneous cap-binding complex-UNK-PABPC interaction is at all possible? If that is actually the main subject of the follow-up publication I would be satisfied with a more general indication.

Indeed, how UNK regulates ribosome occupancy via its effector interface remains an important open question. We previously found using sucrose density fractionation that a substantial fraction of endogenous UNK cosediments with transcripts associated with the small ribosomal subunit (Murn et al, 2015, PMID: 25737280). This suggests that UNK might reduce ribosome occupancy of its targeted mRNAs, at least in part, by interfering with translational initiation. In agreement with the Reviewer's speculation, our current mass spec analysis identified in the UNK complex several translation initiation factors, including eIF4G1, eIF4B, eIF3A, and eIF3A, none of which was detected in the mock IP experiment (Supplementary Table 2). Our preliminary co-IP/western experiments with UNK_{WT} and UNK_{dPAM2} further indicate that the UNK-eIF interactions may be mediated by PABPC, although we have not yet rigorously examined this possibility or potential simultaneous interactions of UNK with PABPC and eIFs. It is conceivable that PABPC-dependent UNK-eIF interactions enforce the translational repression by facilitating, in conjunction with CCR4-NOT, formation of a molecular-scale condensate with restricted access of ribosomes. We are currently investigating this possibility.

Minor points:

Publication title -> might be better to actually include UNK in the title. I understand that the general concept needs to be promoted but the specific case is also important.

We have modified the title of the manuscript to "Regulation by the RNA-binding protein Unkempt at its effector interface".

Fig. 1F -> I find the y-axis somewhat unclear. And would therefore ask for clarification in the caption or on the main figure: What is the exact difference between samples marked 1, 2, 3... 11, 12, 13, 14, 15 etc. and UNK(d11), UNK(d14)? I imagine it is (1.) the fragment alone or (2.) the entire protein with the indicated fragment deleted? If so the greek symbol would be best.

That is correct. Samples named by numbers only (1 through 18) in the original Fig. 1F (new Supplementary Fig. 2D) are lysates of cells expressing Gal4-tagged fragments of UNK. Samples named UNK(d11), UNK(d14) etc. (modified to UNK(Δ 11), UNK(Δ 14) etc. in the revised manuscript) are from cells expressing Gal4-tagged full-length UNK protein with the indicated fragment deleted. We clarified this in the legend to Supplementary Fig. 2D in the revised manuscript.

Fig. 1F and 1G -> match the order of samples for better clarity.

We corrected this as suggested (Supplementary Fig. 1D, E in the revised manuscript).

Section title 'Unkempt interacts directly with multiple CCR4-NOT complex subunits' - is somehow confusing. Maybe better to make it more precise? - 'Unkempt interacts directly with CCR4-NOT complex core NOT module and CNOT9, but not with the catalytic module?'

We modified the section title to "Unkempt interacts directly with the NOT and NOT9 modules of the CCR4-NOT complex", to make it more precise, as suggested.

THERE IS NO MESSAGE TO THE EDITOR THAT IS HIDDEN FROM THE AUTHORS.

Reviewer #3 (Remarks to the Author):

How RBPs select and regulate target mRNAs is a difficult problem due to many reasons including transient binding, involvement of IDRs, and ambiguous binding specificity. The authors chose an RBP (Unkempt) that regulates target mRNA by translational inhibition and is crucial for cellular polarization and performed extensive biochemical study, structure prediction, and systems analyses to understand its molecular mechanism of target selection. The biochemical study using various constructs and mutants of Unkempt and effector proteins and AlphaFold-predicted structural models is impressive. The proposed binding model of Unkempt and effectors makes sense. Although the

issue of the transcriptional activity was not clarified and the target selection mechanism was not completely explained by the current binding model, the study substantially improved our understanding of how RBPs could work on target RNA. I have only a few minor comments, but they need to be addressed before publication.

We thank the Reviewer for the encouraging assessment and helpful comments, which we address below.

1. This study focuses on a single RBP. No evidence is provided to claim a general mechanism for RBPs. This study could be a framework for the future studies on the functional mechanisms of other RBPs but not for understanding them. The authors need to avoid generalization through Abstract and Main text.

We agree. To address this comment, we removed several generalizations from the manuscript and, where appropriate, replaced them with narrative that discusses the significance of our findings for regulation by UNK. This includes changing the title of the manuscript from "A paradigm for regulation at the effector interface with RNA-binding proteins" to "Regulation by the RNA-binding protein Unkempt at its effector interface".

2. The authors need to consider moving the study on transcriptional activity to the back and adding discussion why transcriptional and translational activities are difficult to dissect and how they would be coupled. Would it be possible that Unkempt-bound mRNA is transported back into the nuclear?

We now show all results related to the transcriptional activity of UNK as supplements (Supplementary Fig. 2 and 3).

It remains unclear whether endogenous UNK is transcriptionally active in cells. Our consistent failure despite extensive efforts to detect such activity may have been due to a number of reasons, including low amount of nuclear UNK (Supplementary Fig. 1 and Murn et al, 2015, PMID: 25737280), potentially very transient or indirect interactions with chromatin, regulation of a low number of genes, or absence of such activity altogether. Assuming that transcriptional regulation by endogenous UNK nevertheless exists in cells, it might seem counterintuitive that an RBP would combine potent transcriptional activation and translational repression. This could, however, be rationalized in different ways. For instance, UNK could be transcriptionally inducing only a small subset of genes but translationally repressing a much broader cohort of (different) genes. Alternatively, UNK could conceivably activate transcription of a particular protein-coding gene and stay bound to the resulting transcript while keeping it translationally silent until release for localized translation, for instance, at the growth cones of polarizing cells. It remains to be determined whether any of these mechanisms, or a potential import of UNK-bound mRNAs into the nucleus, as proposed by the Reviewer, is operational in cells. We included these comments in the discussion in the revised manuscript (*lines 448-456*).

3. The authors need to describe what d14, d11, d15, and d17 indicates in the main text and figure legend 1. I assumed that they indicate the deletion of constructs 14, 11, 15, and 17.

That is correct, labels d14, d11, d15, and d17 (revised to $\Delta 14$, $\Delta 11$, $\Delta 15$, and $\Delta 17$) indicate deletions of fragments 14, 11, 15, and 17 from the full-length UNK protein. We clarified this in the legend to Supplementary Fig. 2D in the revised manuscript.

4. Why negatively charged residues were chosen for the mutations V511E, I515E and L522E? These changes may have an effect on RNA-binding through the interaction with neighboring positively charged residues.

For our protein-protein interaction studies of UNK with the NOT9 module we immobilized the UNK_{IDR} in pulldown assays in the absence of RNA. AlphaFold predicted an interaction between a helical element in the UNK_{IDR} and the RNA binding groove of the CNOT9 subunit. Inspection of the predicted interface suggested that small hydrophobic residues supported the affinity of this interaction and that mutation to other hydrophobic residues such as alanine may not weaken the interaction sufficiently to lead to an observable effect. Therefore, we chose to introduce a bulkier, charged residue for mutagenesis. We used this strategy in a previous study with a different CNOT9 interactor (see PMID: 34887419). We appreciate the Reviewer's comment and agree that our chosen mutation would not be appropriate for studies involving RNA without additional controls.

5. line 220, this sentence is unclear.

To clarify, we revised the text as follows:

- Original text: Deletion of the UNK's PAM2-like motif, rendering the UNK_{dPAM2} mutant, released from the UNK complex PABPC but not CCR4-NOT, as assessed by mass spectrometry and confirmed by co-IP/western analysis (Figures 3C, 3H, and S2A).

- Edited text: We deleted the identified PAM2-like motif from the full-length UNK and named the resulting mutant protein UNK_{dPAM2}. As assessed by mass spectrometry and confirmed by co-IP/western analysis, UNK_{dPAM2} bound the CCR4-NOT complex, akin to UNK_{WT}, but did not interact with PABPC (Fig. 2C, H, Supplementary Fig. 4A, Supplementary Table 2) (*lines 150-154*).

6. line 302, What is "d positions?"

We are referring to the fourth position of residues in the heptad repeats that form canonical coiled-coil motifs (**Fig. R2**, copied from Truebestein & Leonard, 2016, PMID: 27492088). Hydrophobic residues appear at the *a* and *d* positions and repeat every seven residues; we replaced a subset of these with negatively charged residues such that they would electrostatically repulse and disrupt the coiled-coil interface. We have edited the text to clarify our meaning.

Original text: To validate the dimerization interface, we substituted hydrophobic residues in *d* positions of the coiled-coil for glutamates...

Edited text: To validate the dimerization interface, we substituted hydrophobic residues in *d* positions of the heptad repeats that form coiled-coil motifs for glutamates... (*lines 201, 202*).

Fig. R2. The canonical coiled-coil is characterized by a heptad repeat in which hydrophobic residues are conserved at positions *a* and *d*. Knobs-into-holes packing of two parallel, supercoiled helices results in *a* and *d* layers (source: Truebestein & Leonard, 2016, PMID: 27492088).

REVIEWERS' COMMENTS

Reviewer #1 (Remarks to the Author):

In the revised manuscript, the authors comprehensively and satisfactorily addressed all my comments as well as other reviewers' points. I think the revised MS is acceptable as it is.

Reviewer #2 (Remarks to the Author):

As Reviewer 2 I previously commented on the work by Shah et al., which is now entitled 'Regulation by the RNA-binding protein Unkempt 1 at its effector interface'. I uphold my previous statement that the work is of interest for publication in Nature Communications.

The authors responded reasonably to all the biological questions in the rebuttal letter. Graphical changes to figures were also included either in the main panels or in the supplement. Overall I think that the authors have positively revised the manuscript. I congratulate them on the work and wish a successful follow-up.

I do have some minor comments, which are only to the discretion of the authors:

(1) Figure 1 has been strongly mutilated and I don't really think that this is an improvement. I would consider to at least move Figures S2E-F to the main panel.

(2) Revised Figures S8B-D present the scatterplot I requested. Those show changes in ribosome occupancy to fold-change in mRNA level. One can clearly see that there is a correlation between the decrease in ribosome occupancy and the decrease in mRNA levels. However the change in ribosome occupancy is more important than the change in mRNA level. This does save the authors main statements in the relevant part of the text (i.e. the original caveat to data interpretation being that UNK regulates mRNA decay and the observed decrease in ribosome occupancy is a consequence of, and proportional to mRNA down-regulation). Graphically this conclusion would be even better shown if a linear trend line for the scatter data was inserted to be compared to the diagonal that is already present. I would prefer to see this data in the main section, but it is the decision of the authors if they want to leave it in the supplement. Clearly ribosome occupancy changes should be first normalized to fold-change in mRNA level before any more detailed conclusions can be drawn in the future.

There are no comments to the editor only.

Reviewer #3 (Remarks to the Author):

The authors addressed all my concerns. I have no more comment and recommend for publication of the revised manuscript.

REVIEWERS' COMMENTS

Reviewer #1 (Remarks to the Author):

In the revised manuscript, the authors comprehensively and satisfactorily addressed all my comments as well as other reviewers' points. I think the revised MS is acceptable as it is.

We took no further action.

Reviewer #2 (Remarks to the Author):

As Reviewer 2 I previously commented on the work by Shah et al., which is now entitled 'Regulation by the RNA-binding protein Unkempt 1 at its effector interface'. I uphold my previous statement that the work is of interest for publication in Nature Communications.

The authors responded reasonably to all the biological questions in the rebuttal letter. Graphical changes to figures were also included either in the main panels or in the supplement. Overall I think that the authors have positively revised the manuscript. I congratulate them on the work and wish a successful follow-up.

I do have some minor comments, which are only to the discretion of the authors:

(1) Figure 1 has been strongly mutilated and I don't really think that this is an improvement. I would consider to at least move Figures S2E-F to the main panel.

When we first revised the manuscript, we modified Figure 1 to show only the essential information gleaned from the study of transcriptional activity, while moving the details to the supplements, as requested by Reviewer #3, point 2. Moving some parts of the transcriptional study (like the proposed Fig. S2E, S2F) but not others back to the main figures could make the flow of the story difficult to follow and would counter the above request by Reviewer #3. We thus decided to keep the entire transcriptional part of the study as Supplementary Information.

(2) Revised Figures S8B-D present the scatterplot I requested. Those show changes in ribosome occupancy to fold-change in mRNA level. One can clearly see that there is a correlation between the decrease in ribosome occupancy and the decrease in mRNA levels. However the change in ribosome occupancy is more important than the change in mRNA level. This does save the authors main statements in the relevant part of the text (i.e. the original caveat to data interpretation being that UNK regulates mRNA decay and the observed decrease in ribosome occupancy is a consequence of, and proportional to mRNA down-regulation). Graphically this conclusion would be even better shown if a linear trend line for the scatter data was inserted to be compared to the diagonal that is already present. I would prefer to see this data in the main section, but it is the decision of the authors if they want to leave it in the supplement. Clearly ribosome occupancy changes should be first normalized to fold-change in mRNA level before any more detailed conclusions can be drawn in the future.

We retained the scatterplots in Fig. S8B-D as Supplementary Information for two reasons. First, these plots do not integrate the UNK-RNA binding information (iCLIP data), which is critical for appreciation of the directness of regulation by UNK. In contrast, the volcano plots in Fig. 6A-6G include the RNA-binding information and show that the observed changes in mRNAs levels only poorly correlate with UNK-mRNA binding (indicating largely indirect effects; see also our reply to point 5 initially raised by Reviewer #1), whereas the translational repression is overwhelmingly a direct consequence of UNK-mRNA interactions. Second, via largely indirect effects, UNK significantly alters steady-state levels of several thousand mRNAs, which in the scatterplots in Fig. S8B-D masks its potent, direct repressive effect on translation. Because the slope of a linear trend line would be heavily affected by the indirect effects of UNK, we chose to omit the trend lines from these scatterplots.

Reviewer #3 (Remarks to the Author):

The authors addressed all my concerns. I have no more comment and recommend for publication of the revised manuscript.

We took no further action.